# Discriminative Calibration: Check Bayesian Computation from Simulations and Flexible Classifier

**Yuling Yao**
Flatiron Institute,
New York, NY 10010.
yyao@yyao.dev

**Justin Domke**
University of Massachusetts,
Amherst, MA 01002.
domke@cs.umass.edu

## Abstract

To check the accuracy of Bayesian computations, it is common to use rank-based simulation-based calibration (SBC). However, SBC has drawbacks: The test statistic is somewhat ad-hoc, interactions are difficult to examine, multiple testing is a challenge, and the resulting p-value is not a divergence metric. We propose to replace the marginal rank test with a flexible classification approach that learns test statistics from data. This measure typically has a higher statistical power than the SBC test and returns an interpretable divergence measure of miscalibration, computed from classification accuracy. This approach can be used with different data generating processes to address simulation-based inference or traditional inference methods like Markov chain Monte Carlo or variational inference. We illustrate an automated implementation using neural networks and statistically-inspired features, and validate the method with numerical and real data experiments.

## 1 Introduction

Simulation based calibration (SBC) is a default approach to diagnose Bayesian computation. SBC was originally designed to validate if computer software accurately draws samples from the exact posterior inference, such as Markov chain Monte Carlo (MCMC, [4, 38, 26]) and variational inference [41]. With recent advances in amortized and simulation-based inferences [6] and growing doubt on the sampling quality [23, 18], there has also been an increasing trend to apply SBC to likelihood-free inference such as approximate Bayesian computation [42] and normalizing-flow-based [31, 1] neural posterior sampling [22, 20], with a wide range of applications in science [15, 7, 34].

Bayesian computation tries to sample from the posterior distribution $p(\theta|y)$ given data $y$. We work with the general setting where it may or may not be possible to evaluate the likelihood $p(y|\theta)$. Suppose we have an inference algorithm or software $q(\theta|y)$ that attempts to approximate $p(\theta|y)$, and we would like to assess if this $q$ is calibrated, meaning if $q(\theta|y) = p(\theta|y)$ for all possible $\theta, y$. Simulation based calibration involves three steps: First, we draw a $\theta$ from the prior distribution $p(\theta)$. Second, we simulate a synthetic observation $y$ from the data model $p(y|\theta)$. Third, given $y$ we draw a size-$M$ posterior sample $\tilde\theta_1, \ldots, \tilde\theta_M$ from the inference engine $q(\theta|y)$ that we need to diagnose. SBC traditionally computes the rank statistic of the prior draw $\theta$ among the $q$ samples, i.e. $r = \sum_{m=1}^{M} \mathbb{1}(\theta \le \tilde\theta_m)$. If the inference $q$ is calibrated, then given $y$ both $\theta$ and $\tilde\theta_m$ are from the same distribution $p(\theta|y)$, hence with repeated simulations of $(\theta, y)$, we should expect such rank statistics $r$ to appear uniform, which can be checked by a histogram visualization or a formal uniformity test.

Despite its popularity, rank-based SBC has limitations: (i) We only compute the *rank* of univariate parameters. In practice, $\theta$ and $y$ are high dimensional. We typically run SBC on each component of $\theta$ separately, this creates many marginal histograms and does not diagnose the joint distribution or interactions. We may compare ranks of some one-dimensional test statistics, but there is no method to

37th Conference on Neural Information Processing Systems (NeurIPS 2023).

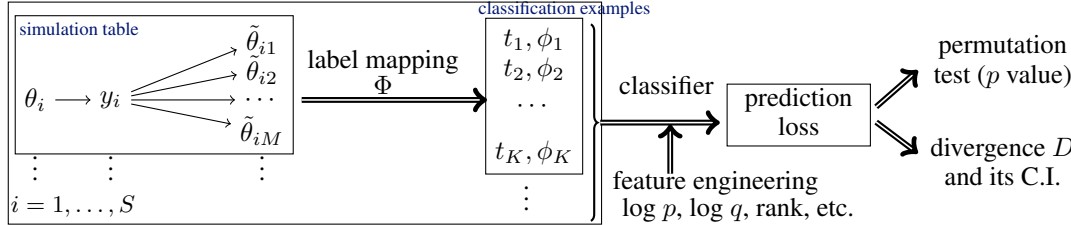

Figure 1: *Our discriminate calibration framework has three modules (a) generate simulation table $(\theta, y, \tilde{\theta})$, and map it into classification examples with some label $t$ and feature $\phi$, (b) train a classifier to predict labels, (c) from the learned classifier, perform hypothesis testing and estimate a divergence.*

find the best summary test statistic. (ii) As long as we test multiple components of $\theta$ or test statistics, directly computing the uniformity $p$-values is invalid (subject to false discovery) unless we make a *multiple-testing* adjustment, which drops the test power (subject to false negative) in high dimensions. (iii) Often we know inference is approximate, so the final goal of diagnostic is not to reject the null hypothesis of perfect calibration but to measure the *degree of miscalibration*. The $p$-value is not such a measure: neither can we conclude an inference with $p = .02$ is better than $p = .01$, nor connect the $p$-value with the posterior inference error. The evidence lower bound, a quantity common in variational inference, also does not directly measure the divergence due to the unknown entropy.

**Heuristic: calibration via classification.** To address all these drawbacks, while maintaining versatility in various computing tasks, we propose *discriminative calibration*, a pragmatic and unified framework for Bayesian computing calibration and divergence estimation. The intuitive heuristic behind discriminative calibration is to use a classifier to perform similarity tests—when two or several distributions are similar, we cannot distinguish samples from them so the classification error is large. In Bayesian computation, we compare conditional distributions, the true $p(\theta|y)$ and inferred $q(\theta|y)$, to which we have access via simulations, and in some occasions explicit densities. It is natural to try to classify the samples drawn from $p$ v.s. from $q$, and the ability to distinguish them would suggest a miscalibration between $p$ and $q$.

To formulate such heuristics into an algorithm, we design a family of "label mapping" that prepares the simulations into classification examples that contain the label and feature. In Sec. 2, we first give four concrete label mapping examples, where the rank-based SBC becomes essentially a special case. Sec. 3 states the general theory: if we train a classifier to predict the labels, then the prediction ability yields a computable divergence from $p(\theta|y)$ to $q(\theta|y)$. In Sec. 4, we illustrate the practical implementation to get the divergence estimate, its confidence interval, and a valid hypothesis testing $p$-value. We explain why the learned classifier helps statistical power. In Sec. 4.1, we discuss the classifier design and incorporate extra known information such as ranks, likelihood, and log densities, whenever available, as features. We also show our method is applicable to MCMC without waste from thinning. We illustrate numerical and cosmology data examples in Sec. 5. We review other related posterior validation approaches and discuss the limitation and future direction in Sec. 6.

## 2 Generate labels, run a classifier, and obtain a divergence

As with traditional SBC, we generate a simulation table by repeatedly sampling parameters $\theta$ and synthetic data $y$ from the target model $p$, i.e. draws $(\theta, y) \sim p(\theta, y)$. Then, for each $(\theta, y)$, we run the inference routine $q$ to obtain a set of $M$ IID approximate posterior samples $\tilde{\theta}_1, \cdots, \tilde{\theta}_M \sim q(\theta|y)$. We wish to assess how close, on average (over different $y$), the inference procedure $q(\theta|y)$ is to the true posterior $p(\theta|y)$. Here, we observe that classification example-sets can be created in several ways and these produce different divergences. We generalize the four examples and the claims on divergence in Sec. 3.

**Input**:
one simulation run:
$(y, \theta, \tilde{\theta}_1, \ldots, \tilde{\theta}_M)$;
**Output**:
$M + 1$ examples:

| label $t$ | features $\phi$ |
|-----------|-----------------|
| 0 | $(\theta, y)$ |
| 1 | $(\tilde{\theta}_1, y)$ |
| 1 | $\cdots$ |
| 1 | $(\tilde{\theta}_M, y)$ |

**Example 1: Binary classification with full features.** An intuitive way to estimate this closeness between $q(\theta|y)$ and $p(\theta|y)$ is to train a binary classifier. Imagine creating a binary classification dataset of $(t, \phi)$ pairs, where $t$ is a binary label, and $\phi$ are features. For each $(\theta, y)$ simulated from $p$, $M + 1$ pairs of examples are created. In the first, $t = 0$ and $\phi = (\theta, y)$. In the others, $t = 1$, and $\phi = (\tilde{\theta}_m, y)$, $1 \le m \le M$. Collecting all data across

$1 \leq i \leq S$, we obtain $S(M+1)$ pairs of $(t, \phi)$ classification examples. A binary classifier is then trained to maximize the conditional log probability of $t$ given $\phi$. If inference $q$ were exact, no useful classification would be possible. In that case, the expected test log predictive density could be no higher than the negative binary entropy $h(w) := w \log w + (1-w) \log(1-w)$ of a Bernoulli distribution with parameter $w := 1/(M+1)$.

Now imagine drawing a validation set in the same way, and evaluating the log predictive density of the learned classifier. We will show below (Thm. 1) that the expected log predicted density [10] $\mathrm{ELPD} = \mathbb{E} \log \Pr(t|\phi)$ of the classifier on validation data is a lower bound to a divergence $D_1$ between $p(\theta|y)$ and $q(\theta|y)$ up to the known constant $h(w)$,

$$\mathrm{ELPD}_1 - h(w) \leq D_1(p, q) := w\mathrm{KL}\left(p(\theta|y) \parallel r(\theta|y)\right) + (1-w)\mathrm{KL}\left(q(\theta|y) \parallel r(\theta|y)\right), \quad (1)$$

where $w = 1/(M+1)$ and $r(\theta|y) = wp(\theta|y) + (1-w)q(\theta|y)$ is a mixture of posterior density. If the classifier $c$ is optimal ($c(t|\phi) = \Pr(t|\phi)$ in the distribution) then the bound in the above equation is tight $\max_{\text{classifiers}} \mathrm{ELPD}_1 - h(w) = D_1(p, q)$. Here $\mathrm{KL}(p(\theta|y) \| q(\theta|y))$ denotes a standard conditional Kullback–Leibler divergence[1]. By optimizing the classifier, $\max \mathrm{ELPD}_1 - h(w)$ becomes a commutable divergence, and its approaching zero is a necessary and sufficient condition for perfect calibration since $D_1(p, q) = 0$ if and only if $p(\theta|y) = q(\theta|y)$ almost everywhere.

**Example 2: Binary classification without $y$.** Similar to Example 1, from each simulation draw we generate $M+1$ pairs of $(t, \phi)$ examples, except that the feature $\phi$ only contains the parameters $\theta$, not $y$. A binary classifier is then trained to predict $t$ given $\phi$. The ELPD of this classifier on validation data is a lower bound to a generalized divergence $D_2$ between the prior $p(\theta)$ and $q(\theta)$, up to a known constant $h(w)$

| $t$ | $\phi$ |
|---|---|
| 0 | $\theta$ |
| 1 | $\tilde{\theta}_1$ |
| 1 | $\cdot$ |
| 1 | $\tilde{\theta}_M$ |

$$\mathrm{ELPD}_2 - h(w) \leq D_2(p, q) := w\mathrm{KL}\left(p(\theta) \parallel r(\theta)\right) + (1-w)\mathrm{KL}\left(q(\theta) \parallel r(\theta)\right), \quad (2)$$

where $w = (M+1)^{-1}$, $r(\theta) = wp(\theta) + (1-w)q(\theta)$ is the prior mixture, and the bound is tight when the classifier is optimal. A large ELPD reveals the difference between the inference $q(\theta|y)$ and $p(\theta|y)$. But $D_2$ is only a generalized divergence: $D_2 = 0$ is necessary not sufficient for $q(\theta|y) = p(\theta|y)$.

**Example 3: Binary classification with ranks (where the classical SBC is a special case).** Instead of classification using full $(\theta, y)$, we construct a *feature*: the rank statistics. From each simulation draw we generate $M+1$ pairs of $(t, \phi)$ examples. The first pair is $t = 0$ and $\phi = \sum_{m=1}^{M} \mathbb{1}(\theta < \tilde{\theta}_m)$, the rank statistics of the prior draw. In the others, $t = 1$ and $\phi = \sum_{m'=1}^{M} \mathbb{1}(\tilde{\theta}_m < \tilde{\theta}_{m'}) + \mathbb{1}(\tilde{\theta}_m < \theta)$, $1 \leq m' \leq M$ are the rank statistics of the inferred samples. A binary

**Output**: $M+1$ examples:

| $t$ | $\phi$ |
|---|---|
| 0 | $\sum_{m=1}^{M} \mathbb{1}(\theta \leq \tilde{\theta}_m)$ |
| 1 | $\sum_{m'=1}^{M} \mathbb{1}(\tilde{\theta}_1 \leq \tilde{\theta}_{m'})$ |
| 1 | $\cdots$ |
| 1 | $\sum_{m'=1}^{M} \mathbb{1}(\tilde{\theta}_M \leq \tilde{\theta}_{m'})$ |

classifier is then trained to predict $t$ given $\phi$. The ELPD of this classifier is a lower bound to a generalized divergence $D_3$ between $p(\theta|y)$ and $q(\theta|y)$ up to a known constant $h(w)$

$$\mathrm{ELPD}_3 - h(w) \leq D_3(p, q) := D_2(Z(p, q) \parallel \mathrm{Uniform}(0, 1)), \quad w = 1/(M+1), \quad (3)$$

and again the bound is tight if the classifier is optimal. Here $Z(p, q)$ is a random variable defined by $Z = Q(\theta|y), (\theta, y) \sim p(\theta, y)$, where $Q$ is the cumulative distribution function of $q(\theta|y)$.

Training a "generative" classifier on this rank-based label-generating map is similar to testing for uniformity in rank statistics, as done in traditional SBC which estimates the distribution of $r|t = 0$ by histograms (See Appendix A.1 for precise correspondence between SBC and the naive Bayes classifier). The success of SBC suggests the usefulness of ranks, which motivates us to include ranks or more generally feature engineering in the classifier. However, $D_3$ is only a generalized divergence: $D_3 = 0$ is necessary but not sufficient for $p(\theta|y) = q(\theta|y)$. If inference always returns the prior, $q(\theta|y) = p(\theta)$, then $D_3(p, q) = 0$, a known counterexample of when rank-based SBC fails [32].

**Example 4: Multi-class classification.** We go beyond binary labeling. Given a simulation run $(y, \theta, \tilde{\theta}_1, \cdots, \tilde{\theta}_M)$, we now create an $(M+1)$-class classification dataset with $M+1$ pairs of $(t, \phi)$ examples. In each one, the features are $\phi = (y, \theta^*)$ where $\theta^*$ is a permutation of $(\theta, \tilde{\theta}_1, \cdots, \tilde{\theta}_M)$ that moves $\theta$ into a given location and $t \in 0, \cdots, M$ indicates the location of $\theta$ in the permutation

**Output**: $M+1$ examples:

| $t$ | $\phi$ |
|---|---|
| 0 | $(\theta, \tilde{\theta}_1, \tilde{\theta}_2, \cdots, \tilde{\theta}_M, y)$ |
| 1 | $(\tilde{\theta}_1, \theta, \tilde{\theta}_2 \cdots, \tilde{\theta}_M, y)$ |
| $\cdot$ | $\cdots$ |
| $M$ | $(\tilde{\theta}_1, \tilde{\theta}_2 \cdots, \tilde{\theta}_M, \theta, y)$ |

---

[1]Standard notation for conditional divergences [5] is that $\mathrm{KL}(p(\theta|y)\|q(\theta|y)) := \mathbb{E}_{p(y,\theta)} \log \frac{p(\theta|y)}{q(\theta|y)}$. Conditional divergence is *not* the divergence of conditional distributions. We interpret $p(y) = q(y)$.

(See the table on the right). We train a $(M+1)$-class classifier to predict $t$ from $\phi$. The ELPD on a validation set is a lower bound to the following divergence $D_4$ between $p(\theta|y)$ and $q(\theta|y)$ up to a known constant:

$$\text{ELPD}_4 + \log(M+1) \leq D_4(p, q) \coloneqq \text{KL}\left(p(\theta_0)\prod_{k=1}^{M} q(\theta_k), \ \frac{1}{M+1}\sum_{m=0}^{M} p(\theta_m)\prod_{k \neq m} q(\theta_k)\right). \quad (4)$$

Again the bound is tight if the classifier is optimal. The divergence $0 \leq D_4 \leq \log(M+1)$, and $D_4 = 0$ if and only if $p(\theta|y) \overset{\text{a.e.}}{=} q(\theta|y)$, necessary and sufficient for calibration. In Theorem 3 we shows that as $M \to \infty$, $D_4(p, q)$ converges to $\text{KL}(p(\theta|y), q(\theta|y))$ at an $O(1/M)$ convergence rate.

## 3 Theory on calibration divergence

To generalize the previous examples, we define a "**label mapping**" $\Phi : (y, \theta, \tilde{\theta}_1, \ldots, \tilde{\theta}_M) \mapsto \{(t_1, \phi_1), \ldots, (t_L, \phi_L)\}$. that maps one simulation run $(y, \theta, \tilde{\theta}_1, \ldots, \tilde{\theta}_M)$ into a $K$-class classification example-set containing $L$ pairs of labels $t$ and features $\phi$. The label $t_l \in \{0, 1, \ldots, K-1\}$ is deterministic and only depends on $l$. The features $\phi_l$ can depend on $l$ and $(y, \theta, \tilde{\theta}_1, \ldots, \tilde{\theta}_M)$. We only consider $\Phi$ satisfying that, when $p(\theta|y) = q(\theta|y)$, $\phi$ given $y$ is conditionally independent of $t$ (equivalently, $\phi|(y, t)$ has the same distribution for any $t$). Let $\mathbb{F}$ be the set of these mappings $\Phi$.

We train a classifier on the classification examples created by $\Phi$ collected across repeated sampling. The classifier performance is measured by its expected log predictive density (ELPD), or equivalently the negative cross-entropy loss. Given $p$, $q$, and the mapping $\Phi$, let $c(\phi)$ be any probabilistic classifier that uses $\phi$ to predict the label $t$, and $\Pr(k|\phi, c)$ is the predicted $k$-th class probability. Taking expectations over features and labels with $(\theta, y) \sim p$ and $\tilde{\theta}_m \sim q(\theta|y)$ reaches the ELPD of the classifier,

$$\text{ELPD}(\Phi, c) \coloneqq \mathbb{E}_{t, \phi} \log \Pr(t = k|\phi, c), \quad (t, \phi) = \Phi(y, \theta, \tilde{\theta}_1, \ldots, \tilde{\theta}_M) \quad (5)$$

We then define the **prediction ability** $D(p, q, \Phi, c)$ to be the ELPD plus the entropy of a categorical distribution, i.e.

$$D(p, q, \Phi, c) = \text{ELPD}(\Phi, c) - \sum_{k=0}^{K-1} w_k \log w_k, \quad \text{where } w_k = \frac{1}{L}\sum_{l=1}^{L} \mathbb{1}(t_l = k). \quad (6)$$

The optimal classifier is the $c$ that achieves the highest prediction ability in the population:

$$D^{\text{opt}}(p, q, \Phi) \coloneqq \max_{c \in \mathcal{C}} D(p, q, \Phi, c), \quad \text{where } \mathcal{C} \text{ is the set of all probabilistic classifiers.} \quad (7)$$

The next theorem is the basic theory of our method: as long as we pass the simulation draws to a label mapping $\Phi \in \mathbb{F}$, and train a classifier on the classification examples, then $D^{\text{opt}}(p, q, \Phi)$ is a generalized divergence between $p$ and $q$.

**Theorem 1** (Prediction ability yields divergence). *Given any $p, q$, and feature mapping $\Phi \in \mathbb{F}$, the optimal prediction ability $D^{\text{opt}}(p, q, \Phi)$ is a generalized divergence from $p$ to $q$ in the sense that $D^{\text{opt}}(p, q, \Phi) \geq 0$, and $p(\theta|y) = q(\theta|y)$ almost everywhere implies $D^{\text{opt}}(p, q, \Phi) = 0$. This generalized divergence is reparametrization invariant and uniformly bounded. For any classifier $c$,*

$$0 \leq D(p, q, \Phi, c) \leq D^{\text{opt}}(p, q, \Phi) \leq -\sum_{k=0}^{K-1} w_k \log w_k; \quad (8)$$

$$p(\theta|y) \overset{\text{a.e.}}{=} q(\theta|y) \Rightarrow D^{\text{opt}}(p, q, \Phi) = 0.$$

That is, any label mapping $\Phi$ produces a generalized divergence $D^{\text{opt}}(p, q, \Phi)$, the prediction ability of the optimal classifier. The prediction ability $D(p, q, \Phi, c)$ of any classifier $c$ estimated on validation data is a lower bound to $D^{\text{opt}}(p, q, \Phi)$. Further, this generalized divergence $D^{\text{opt}}(p, q, \Phi)$ is always a proper Jensen–Shannon divergence in the projected feature-label space (Theorem 1 in the Appendix).

When $p(\theta|y) \neq q(\theta|y)$, to increase the statistical power, we wish that the generalized divergence can be as "strong" as possible such that we can detect the miscalibration. In the four examples in Section 2, we have used $D_1, D_2, D_3, D_4$ to denote the (generalized) divergence they yield. The next theorem shows there is a deterministic domination order among these four metrics. Moreover, $D_4$ is the largest possible classification divergence from any given simulation table.

**Theorem 2** (Strongest divergence). *For any given $p, q$, and any $\Phi \in \mathbb{F}$, (1) $D_4 \geq D_1 \geq D_3 \geq D_2$. (2) $D_4$ and $D_1$ are proper divergences. They attain 0 if and only if $p(\theta|y) = q(\theta|y)$ almost everywhere. They attain the corresponding upper bound in (8) if and only $p(\theta|y)$ are $q(\theta|y)$ are disjoint, i.e., $\int_A p(\theta|y)q(\theta|y)d\theta = 0$ for any measurable set $A$ and almost surely $y$. (3) For any $p, q$ and $\Phi \in \mathbb{F}$ ($\Phi$ can have an arbitrary example size $L$ and class size $K$), $D_4(p,q) \geq D^{\mathrm{opt}}(p,q,\Phi)$.*

The following result shows that the divergence $D_4$ in Eq. (4) approaches the "mode-spanning" KL divergence in the limit that the number of posterior draws $M \to \infty$. This is appealing because for many inference routes, increasing the number of posterior draws is cheap. Thus, $D_4$ provides an accessible approximation to the KL divergence that is otherwise difficult to compute from samples.

**Theorem 3** (Big $M$ limit and rate). *For any $p, q$, generate the simulation $\{(y, \theta, \tilde{\theta}_i, \dots, \tilde{\theta}_M)\}$ and train the multiclass-classier, then $D_4(p,q) - \mathrm{KL}(p(\theta|y) \| q(\theta|y)) \to 0$, as $M \to \infty$.*

*If further $p(\theta|y)$ and $q(\theta|y)$ have the same support, and if $\mathbb{E}_{p(\theta|y)} \left[ \frac{p(\theta|y)}{q(\theta|y)} \right]^2 < \infty$ for a.s. $y$, then*

$$D_4(p,q) = \mathrm{KL}(p(\theta|y) \| q(\theta|y)) - \frac{1}{2M}\chi^2(q(\theta|y) \| p(\theta|y)) + o(M^{-1}).$$

*where $\chi^2(\cdot\|\cdot)$ is the conditional chi-squared divergence.*

## 4 Practical implementation

---

**Algorithm 1:** Proposed method: Discriminative calibration

---

**input** : The ability to sample from $p(\theta, y)$ and $q(\theta|y)$, and a label mapping $\Phi$.
**output :**(i) estimate of a divergence between $p(\theta|y)$ and $q(\theta|y)$; (ii) $p$-value for testing $p(\theta|y) = q(\theta|y)$.
**for** ( $i = 1 : S$ )

   Sample $(\theta_i, y_i) \sim p(\theta, y)$, and sample $\tilde{\theta}_{i1}, \tilde{\theta}_{i2} \dots, \tilde{\theta}_{iM} \sim q(\theta|y_i)$;       ▷ simulation table

   Generate a batch of $L$ examples of $(t, \phi) = \Phi(y_i, \theta_i, \tilde{\theta}_{i1}, \dots, \tilde{\theta}_{iM})$, $0 \leq t \leq K - 1$; ▷ label mapping

Randomly split the $LS$ classification examples $(t, \phi)$ into training and validation sets (all $L$ examples for a given $i$ go to either training or validation);

Train a $K$-class classifier to predict label $t$ on the training examples, incorporating useful features;

Compute the validation log predictive density $\mathrm{LPD}_{\mathrm{val}}$ in (9), obtain an estimate of the divergence (7) and its bootstrap confidence intervals;                                ▷ divergence

**for** ( $b = 1 : B$ )

   Randomly permute the label $t$ in the validation set within each batch;

   Compute $\mathrm{LPD}_{\mathrm{val}}^b$ on the permutated validation set;

Compute the calibration $p$-value $p = 1/B \sum_{b=1}^B \mathbb{1}(\mathrm{LPD}_b^{\mathrm{val}} \geq \mathrm{LPD}^{\mathrm{val}})$.         ▷ frequentist test

---

**Workflow for divergence estimate.** We repeat the simulation of $(y, \theta, \tilde{\theta}_1, \dots, \tilde{\theta}_M)$ for $S$ times. Each time we sample $(y, \theta) \sim p(\theta, y)$ and $\tilde{\theta}_{1:M} \sim q(\theta|y)$ and generate a *batch* of $L$ examples through a label mapping $\Phi \in \mathbb{F}$. In total, we obtain $SL$ pairs of $(t, \phi)$ classification examples. We recommend using the binary and multiclass labeling schemes (Examples 1 and 4, where $L = M+1$ and $K = 2$ or $M + 1$) such that we can obtain a proper divergence estimate. We split the classification example-set $\{(t, \phi)\}$ into the training and validation set (do not split batches) and train a $K$-class classifier $c$ on the training set to minimize cross-entropy. Denote $\mathcal{I}_{\mathrm{val}}$ to be the validation index, $\mathrm{Pr}(t = t_j|\phi_j, c)$ to be the learned class probability for any validation example $(t_j, \phi_j)$, we compute the ELPD (5) by the validation set log predictive density:

$$\mathrm{ELPD}(\Phi, c) \approx \mathrm{LPD}^{\mathrm{val}}(\Phi, c) := |\mathcal{I}_{\mathrm{val}}|^{-1} \sum_{j : \in \mathcal{I}_{\mathrm{val}}} \log \mathrm{Pr}(t = t_j|\phi_j, c). \tag{9}$$

For any $c$, $\mathrm{LPD}^{\mathrm{val}}(\Phi, c) - \sum_{k=0}^{K-1} w_k \log w_k$ becomes a lower bound estimate of the (generalized) divergence $D^{\mathrm{opt}}(p, q, \Phi)$ in Thm. 1, and an estimate of $D^{\mathrm{opt}}(p, q, \Phi)$ itself when the classier $c$ is good enough. In addition to the point estimate of the divergence, we can compute the confidence interval. It is straightforward to obtain the standard error of the sample mean (9). To take into account the potentially heavy tail of the log predictive densities, we can also use Bayesian bootstrap [37] that reweights the sum in (9) by a uniform Dirichlet weight.

**Hypothesis testing.** Our approach facilitates rigorous frequentist hypothesis testing. The null hypothesis is that the approximate inference matches the exact posterior, i.e., $p(\theta|y) = q(\theta|y)$ almost everywhere. We adopt the permutation test: We train the classifier $c$ once on the training set and keep it fixed, and evaluate the validation set log predictive density $\text{LPD}^{\text{val}}(\Phi, c)$ in (9). Next, permutate the validation set $B$ times: at time $b$, keep the features unchanged and randomly permutate the validation labels $t$ within each batch of examples ($\Phi$ generates a batch of $L$ examples each time), and reevaluate the validation set log predictive density (9) on permutated labels, call it $\text{LPD}^{\text{val}}_b$. Then we compute the one-sided permutation $p$-value as $p = \sum_{b=1}^{B} \mathbb{1}(\text{LPD}^{\text{val}}_b \geq \text{LPD}^{\text{val}})/B$. Given a significance level, say 0.05, we will reject the null if $p < 0.05$ and conclude a miscalibration.

**Theorem 4** (Finite sample frequentist test). *For any finite simulation size $S$ and posterior draw size $M$, and any classifier $c$, under the null hypothesis $p(\theta|y) = q(\theta|y)$ almost everywhere, the permutation test is valid as the $p$-value computed above is uniformly distributed on $[0, 1]$.*

Our test is exact when the simulation size $S$ and $M$ is finite, while the original SBC [4] relied on asymptotic approximation. Further, we *learn* the test statistic via the classifier, and our test is valid regardless of the dimension of $\theta$ and there is no need to worry about post-learning testing [3], while the usual SBC rank test will suffer from low power due to multiple testing.

Our test is always valid even if the classifier $c$ is not optimal. Why does our *learning* step help? For notional brevity, here we only reason for the binary classification. For any $p, q$, we apply the binary classification as described in Example 1, $t = 0$ or 1 and $\phi = (\theta, y)$.

**Theorem 5** (Sufficiency). *Let $\hat{c}(\theta, y) = \Pr(t = 1|\theta, y)$ be the probability of label 1 in the optimal classifier as per (7), and let $\pi_c^p$ and $\pi_c^q$ be the one-dimensional distributions of this $\hat{c}(\theta, y)$ when $(\theta, y)$ is sampled from $p(\theta, y)$ or from $p(y)q(\theta|y)$ respectively, then (i) Conditional on the summary statistic $\hat{c}$, the label $t$ is independent of features $\phi = (\theta, y)$. (ii) Under regularity conditions, there is no loss of information in divergence as the joint divergence is the same as the projected divergence in the one-dimensional $\hat{c}$-space $D_1(p, q) = D_1(\pi_c^p, \pi_c^q)$.*

That is, the best prediction $\hat{c}$ entails the best one-dimensional summary statistics of the high dimensional $\theta \times y$ space. The enhancement of the test power from using the sufficient statistics is then assured by the Neyman-Pearson lemma [27].

### 4.1 Feature engineering: use rank, log density and symmetry

Reassuringly, whatever the classifier $c$, the prediction ability $D(p, q, \Phi, c)$ is always a lower bound to the corresponding divergence $D^{\text{opt}}(p, q, \Phi)$, and the permutation test is always exact. But we still wish to train a "good" classifier in terms of its out-of-sample performance, for a tighter bound in divergence, and a higher power in testing. We will typically use a flexible parametric family such as a multilayer perceptron (MLP) network to train the classifier.

The oracle optimal probabilistic classifier is the true label probability, and in the binary and multiclass classification (Example 1 and 4), the oracle has closed-form expressions, although we cannot evaluate:

$$\Pr_{\text{binary}}(t = 0|\theta, y) = \frac{p(\theta|y)}{p(\theta|y) + q(\theta|y)}, \quad \Pr_{\text{multi}}(t|\theta_0, \ldots, \theta_M, y) = \frac{p(\theta_t, y)/q(\theta_t|y)}{\sum_{k=0}^{M} p(\theta_k, y)/q(\theta_k|y)}. \quad (10)$$

**Statistically-meaningful feature.** Depending on the inference task, we have more information than just the sample points and should use them in the classifier. In light of the shape and component of the optimal classifiers (10), the following statistically-meaningful features are useful whenever available: (i) The log target density $\log p(\theta|y)$. As proxies, the log joint density $\log p(\theta, y)$ and the log likelihood $\log p(y|\theta)$ are often known to traditional Bayesian models. (ii) The log approximate density $\log q(\theta|y)$, known to volitional and normalizing flows. (iii) When the log approximate density is unknown, a transformation is to integrate the density and obtain the posterior CDF, $Q(\theta|y) = \mathbb{E}_{x \sim q(x|y)} \mathbb{1}(x > \theta)$, and this CDF can be approximated by the rank statistics among the approximate draws up to a rescaling $r(\theta, y) := \sum_{m=1, i=1}^{MS} \mathbb{1}(y_i = y)\mathbb{1}(\theta < \tilde{\theta}_{im})$. See Appendix Table 1 for the usage and extension of these features in binary and multiclass classifiers.

**Linear features.** We call the log likelihood, the log approximate density, or log prior (whenever available) linear features, and denote them to be $l(\theta, y)$. For example if both likelihood and $q$ density

is known, then $l(\theta, y) = (\log p(y|\theta), \log q(\theta|y))$. Because they appear in the oracle 10, we will keep linear features in the last layer of the network (followed by a softmax). We recommend parameterizing the binary classification in the following form:

$$\Pr(t = 1|(\theta, y)) = \text{inv\_logit}[\text{MLP}(\theta, y) + w^T l(\theta, y)]. \tag{11}$$

**Symmetry in multiclass classifier.** The multi-class classifier is harder to train. Luckily, we can use the symmetry of the oracle classifier (10): the probability of class $k$ is proportional to a function of $(\theta_k, y)$ only, hence we recommend parameterizing the multiclass probability as

$$\Pr(t = k|(\theta_0, \theta_1, \ldots, \theta_M, y)) = \frac{\exp(g(\theta_k, y))}{\sum_{k'=0}^{M} \exp(g(\theta_{k'}, y))}, \quad g(\theta, y) = \text{MLP}(\theta, y) + w^T l(\theta, y), \tag{12}$$

where $l(\theta, y)$ is available linear features. We only need to learn a reduced function from $\theta \times y$ to $\mathbb{R}$, instead of from $\theta^{M+1} \times y$ to $\mathbb{R}$, reducing the complexity while still keeping the oracle (10) attainable.

**Zero waste calibration for MCMC.** If $\tilde{\theta}_1, \ldots, \tilde{\theta}_M \sim q(\theta|y)$ are produced by MCMC sampler, typically $\tilde{\theta}$ has autocorrelations. Although classical SBC originated from MCMC application, the rank test requires independent samples, so SBC can only handle autocorrelation by thinning: to subsample $\tilde{\theta}_m$ and hope the thinned $\tilde{\theta}$ draws are independent. Thinning is a waste of draws, inefficient when the simulations are expensive, and the thinned samples are never truly independent. With MCMC draws $\tilde{\theta}$, our method could adopt thinning as our Thm. 1 and 4 are valid even when $M = 1$.

Yet we can do better by using all draws. The "separable" network architecture (12) is ready to use for MCMC samples. For example, we sample $(\theta, y) \sim p(\theta, y)$, and sample $(\tilde{\theta}_1, \ldots, \tilde{\theta}_M)$ from a MCMC sampler whose stationary distribution we believe is $q(\theta|y)$, and generate examples from the multiclass permutation (Example 4). Then we run a separable classifier(12) to predict $t$. Intuitively, the separable network design (12) avoids the interaction between $\tilde{\theta}_m$ with $\tilde{\theta}_{m'}$, and disallows the network to predict $t$ based on the autocorrelation or clustering of $\tilde{\theta}$. The next theorem states the validity of our method in MCMC settings without thinning.

**Theorem 6** (MCMC). *Suppose we sample* $(\theta, y) \sim p(\theta, y)$*, and sample* $(\tilde{\theta}_1, \ldots, \tilde{\theta}_M)$ *from a MCMC sampler whose stationary distribution we believe is* $q(\theta|y)$ *(i.e., marginally* $\tilde{\theta}_i$ *is from* $q(\theta|y)$*), and generate examples* $((t_1, \phi_1), \ldots, (t_{M+1}, \phi_{M+1}))$ *from the multiclass permutation, such that* $\phi = (\theta_0, \theta_1, \ldots, \theta_M)$*. Then we run an exchangeable classifier* (12) *in which g is any* $\Theta \times \mathcal{Y} \to \mathbb{R}$ *mapping. Denote* $D_4^{\text{MCMC,sep}}(p, q)$ *to be the predictive ability of the optimal classifier among all separable classifiers* (12)*, then* $D_4^{\text{MCMC,sep}}(p, q) = D_4(p, q)$*.*

**Dimension reduction and nuisance parameter.** Sometimes we only care about the sampling quality of one or a few key dimensions of the parameter space, then we only need to restrict the classifier to use these targeted dimensions, as a result of Theorem 1. For example, in binary classification, if we reduce the feature $\phi = (\theta, y)$ to $\phi = (h(\theta), y)$ in the classifier, where $h(\theta)$ can be a subset of $\theta$ dimensions, then the resulting classification divergence becomes projected divergence between $h(\theta)|y, \theta \sim p(\theta|y)$ and $h(\theta)|y, \theta \sim q(\theta|y)$, and other nuisance parameters do not impact the diagnostics.

**Weighing for imbalanced binary classification.** When $M$ is big, the binary classification (Example 1) can suffer from imbalanced labels, and the divergence in (1) degenerates: $D_1(p, q) \to 0$ as $M \to \infty$. One solution is to use the multiclass classification which has a meaningful limit (Thm. 3). Another solution is to reweigh the loss function or log predictive density by the label $t$. If the weights of class 1 and class 0 examples are $\frac{M+1}{2M}$ and $\frac{M+1}{2}$, used in both training and validation log prediction density, then regardless of $M$, the weighted classification is equivalent to balanced classification, and the resulting divergence is the symmetric Jensen-Shannon (JS) divergence $\frac{1}{2} \text{KL}[p(\theta|y)||r(\theta|y)] + \frac{1}{2} \text{KL}[q(\theta|y)||r(\theta|y)]$, where $r(\theta|y) = \frac{1}{2}[p(\theta|y) + q(\theta|y)]$. See Appendix A for proofs.

## 5 Experiments

**Closed-form example.** Consider a multivariate normal parameter prior $\theta \in \mathbb{R}^d \sim \text{MVN}(\mathbf{0}, \text{Id}_d)$ and a normal data model $y|\theta \sim \text{MVN}(\theta, \Sigma)$, so the exact posterior $p(\theta|y)$ is explicit. In the experiments,

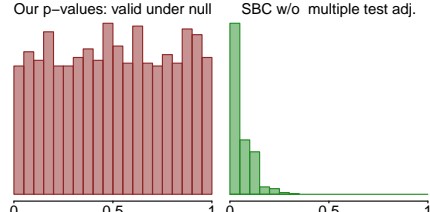

Figure 2: *Our test is valid as it yields uniformly distributed p-values under the null $p(\theta|y) = q(\theta|y)$. We check all dimensions at once.*

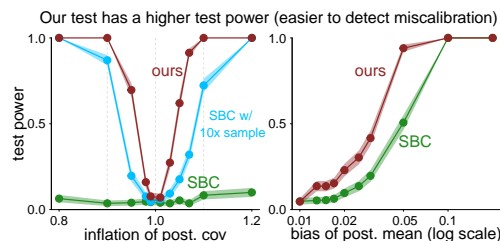

Figure 3: *Our test has a uniformly higher power than SBC rank test. We simulate wrong inference q by multiplying the posterior covariance or adding biases to the posterior mean.*

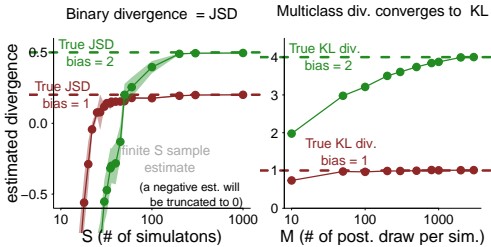

Figure 4: *In the Gaussian example with bias = 1 or 2, with a moderately big simulation sample size S, the divergence estimated from a binary (left panel) or multiclass (right panel) classifier matches its theory quantity (dashed): the Jensen-Shannon or Kullback–Leibler divergence in the big-M limit.*

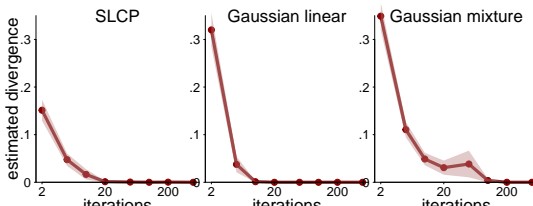

Figure 5: *We apply binary classifier calibration to posterior inferences on three models from the SBI benchmark, and check the sampling quality after some iterations. The x-axis is the number of MCMC iterations. The y-axis is the estimated JS divergence ($\pm$ standard error) between the true target and sampled distribution at a given number of iterations, indicating gradual convergence.*

we use neural nets to train binary (11) or symmetric multiclass classifiers (12), which we generally recommend. We assume the inference posterior $\log q(\theta|y)$ is known and added to the classifier.

*Legitimacy of testing.* First, to validate our hypothesis test under the null, we use true inference $q(\theta|y) = p(\theta|y)$. With $d = 16$, we simulate $S = 500$ draws of $(\theta, y) \sim p(\theta, y)$, then generate true posterior $\tilde{\theta}_m \sim p(\theta|y)$, and run our binary-classifier and obtain a permutation $p$-value. We repeat this testing procedure 1000 times to obtain the distribution of the $p$-values under the null, which is uniform as shown in Fig. 2, in agreement with Thm. 4. Because $\theta$ has $d$ dimensions, a SBC rank test on each margin separately is invalid without adjustment, and would require Bonferroni corrections.

*Power.* We consider two sampling corruptions: (i) bias, where we add a scalar noise (0.01 to 0.2) to each dimension of the true posterior mean, (ii) variance, where we inflate the true posterior covariance matrix by multiplying a scalar factor (0.8 to 1.2). We compare our discriminative tests to a Bonferroni-corrected SBC chi-squared rank-test. In both settings, we fix a 5% type-I error and compute the power from 1000 repeated tests. Fig. 3 shows that our method has uniformly higher power than SBC, sometimes as good as SBC with a 10 times bigger simulation sample size.

*Divergence estimate.* The left panel of Fig. 4 validates Thm. 1: We run a weighted binary classifier on the Gaussian simulations with a scalar bias 1 or 2 added to the posterior mean in $q$. As the number of simulations $S$ grows, the learned classifier quickly approaches the optimal, and the prediction ability matches the theory truth (dashed line): the Jensen-Shannon divergence between $p(\theta|y)$ and $q(\theta|y)$. The right panel validates Thm. 3: With a fixed simulation size $S$, we increase $M$, the number of posterior draws from $q$, and run a multiclass classification. When $M$ is big, the estimated divergence converges from below to the dashed line, which is the theory limit $\mathrm{KL}(p(\theta|y), q(\theta|y))$ in Thm. 3.

**Benchmark examples.** Next, we apply our calibration to three models from the SBI benchmark [23]: the simple likelihood complex posterior (SLCP), the Gaussian linear, and the Gaussian mixture model. In each dataset, we run adaptive No-U-Turn sampler (NUTS) and check the quality of the sampled distribution after a fixed number of iterations, varying from 2 to 2000 (we use equal number of iterations for warm-up and for sampling, and the warm-up samples were thrown away). At each given MCMC iterations, we run our classifier calibration, and estimate the JS divergence, as reported

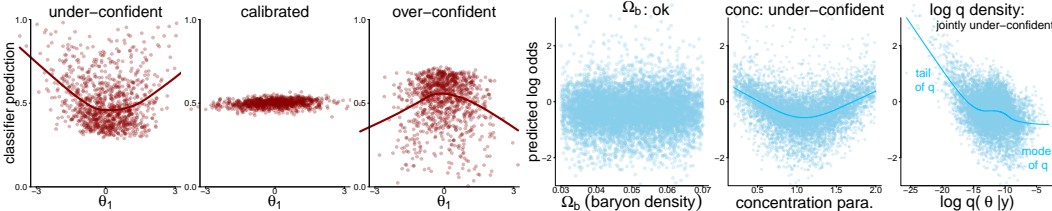

Figure 6: *Scatter plots of the classifier prediction v.s. a one-dimensional parameter we need to check can reveal marginal over- or under-confidence in inference $q(\theta|y)$ from a U or inverted-U shape.*

Figure 7: *In the cosmology example we visually check the classifier predicted log odds v.s. cosmology parameter $\Omega_b$ and* `conc`, *and* $\log q(\theta|y)$. *Underconfidence is detected in* `conc` *marginally, and more so in the joint.*

in Fig. 5. In all three panels, we are able to detect the inference flaws at early iterations and observe a gradual convergence to zero.

**Visual check.** Our diagnostic outputs rigorous numerical estimates, but it also facilities visual checks. We make a scatter plot of the binary classifier prediction $\Pr(t = 1|\phi)$, a proxy of $p(\theta|y)/q(\theta|y)$, against any one-dimensional parameter we need to check: If that parameter is under- or over-confident, this scatter plot will display a U- or inverted-U-shaped trend. Compared with SBC rank histograms, our visualization can further check the magnitude of mismatch (how far away $\Pr(t = 1|\phi)$ is from 0.5), tail behavior (small $q(\theta|y)$), and several dimensions jointly. Fig 6 is a visual check in the Gaussian example when we multiply the posterior covariance in inference $q$ by 0.8, 1, or 1.2.

**Galaxy clustering.** We would like to model galaxy spatial clustering observations in a Lambda

|        | model 1, $d_y$=38 | model 2, $d_y$=151 | model 3, $d_y$=1031 |
|--------|-------------------|--------------------|---------------------|
| 1-flow | 0.015 (0.004)     | 0.076 (0.007)      | 0.068 (0.007)       |
| 5-ens. | 0.013 (0.004)     | 0.044 (0.006)      | 0.035 (0.006)       |

cold dark matter framework [34], where observations $y$ correspond to statistical descriptors of galaxy clustering, and a 14-dimensional parameter $\theta$ encodes key cosmological information about the Universe. The forward simulation $y|\theta$ involves expensive $N$-body simulations (each single $y$ simulation takes around 5000 cpu hours). We consider three different physical models $p$: they contain the same cosmology parameter $\theta$ but three types of observations whose dimension $d_y$ varies from 38 to 1031, reflecting three power-spectrum designs. We apply simulation based inference to sample from each of the models, where we try various normalizing flow architectures and return either the best architecture or the ensemble of five architectures. We now apply our discriminative calibration to these six approximate inferences using a weighted binary classifier trained from $S = 2000, M = 500$ simulations for each inference (that is 1 million examples). We have added the log densities $\log q(\theta|y)$ as extra features in the classifier, as they are known in the normalizing-flows. The table above shows the estimated Jensen–Shannon distances and standard deviation. Compared this table with the estimates from blackbox MLP classifier (Appendix Fig. 8), using statistical features greatly improves the label prediction and tightens the divergence estimate. From the table above, all inferences $q$ have a significantly non-zero but small divergence, among which the 5-ensemble that uses a mixture of 5 flows always has a lower divergence so is more trustworthy. Further visualization of the classification log odds (Fig. 7) reveals that $q$ is under-confident in the parameter that encodes the galaxy concentration rate. To interpret the tail behavior, the predicted log odds are negatively correlated with log joint normalizing-flow density $q(\theta|y)$, suggesting $q(\theta|y) > p(\theta|y)$ in the tail of $q(\theta|y)$, another evidence of $q$ underconfidence.

We share Jax implementation of our binary and multiclass classifier calibration in Github[2].

## 6   Discussions

This paper develops a classifier-based diagnostic tool for Bayesian computation, applicable to MCMC, variational and simulation based inference, or even their ensembles. We learn test statistics from data using classification. Through a flexible label mapping, our method yields a (family of) computable divergence metric and an always valid testing $p$-value, and the statistical power is typically higher than the traditional rank based SBC. Various statically-meaningful features are available depending on the task and we include them in the classifier. Visual exploration is also supported.

---

[2] `https://github.com/yao-yl/DiscCalibration`

**Related diagnostics.** The early idea of simulation based calibration dates back to [13] who compared the prior and the data-averaged posterior, i.e., to check $p(\theta) = \int q(\theta|y)p(y)dy$ using simulations. [42] further developed techniques for the first and second moment estimate of the data-averaged posterior using the law of total variance. Our method includes such moment comparison as special cases: using the no-$y$ binary labeling (Example 2), then comparing the empirical moments of the $q$ sample the $p$ sample can be achieved through a naive Bayes classifier or a linear discriminant analysis.

The rank-based SBC [4, 38] can be recovered by ours using binary labels and taking ranks to be features (Example 3). The rank statistic is central to SBC, which follows the classical frequentist approach—using the tail probability under the posited model quantifies the extremeness of the realized value[11] is only a convenient way to locate the observations in the reference distribution, especially in the past when high dimensional data analysis was not developed—it is the use of modern learning tools that sharpens our diagnostic. Recently, SBC has developed various heuristics for designing (one-dimensional) test statistics $\phi(\theta, y)$. Such rank tests are recovered by our method by including the rank of $\phi$ in our features (Sec. 4.1). For example, [26] proposed to test the rank of the likelihood $\phi = p(\theta|y)$ in MCMC, [20] looked at the rank of proposal density $q(\theta|y)$, [22] used the $q$-probability integral transformation in normalizing flows. In light of our Theorem 5, the optimal test statistic is related to the density ratio and hence problem-specific, which is why our method includes all known useful features in the classifier and learns the test statistic from data.

**Classifier two-sample test.** Using classification to compare two-sample closeness is not a new idea. The classifier two-sample test (C2ST) has inspired the generative adversarial network (GAN) [16] and conditional GAN [25]. [33] has developed GAN-typed inference tools for SBI. In the same vein, [19] used classifiers to diagnose multiple-run MCMC, and [24] used classifiers to learn the likelihood ratio for frequentist calibration, which is further related to using regression to learn the propensity score [36] in observational studies. The theory correspondence between binary classification loss and distribution divergence has been studied in [21, 28, 35]. This present paper not only applies this classifier-for-two-sample-test idea to amortized Bayesian inference to obtain a rigorous test, but also advances the classifier framework by developing the theory of the "label mapping", while the traditional GAN-type approach falls in the one-class-per-group category and deploy binary classifiers as in Example 1. Our extension is particularly useful when samples are overlapped (multiple $\theta$ per $y$), autocorrelated, and imbalanced.

**KL divergence estimate from two samples.** As a byproduct, our proposed multiclass-classifier provides a consistent KL divergence estimate (Thm. 3) from two samples. Compared with existing two-sample KL estimate tools from the $f$-divergence representation [29] or the Donsker-Varadhan representation [2], our multiclass-classifier estimate appears versatile for it applies to samples with between-sample dependence or within-sample auto-correlation. It is plausible to apply our multiclass-classifier approach to other two-sample divergence estimation tasks such as the $f$-GAN [30], which we leave for future investigation.

**Limitations and future directions.** Like traditional SBC, our method assesses the difference between $p(\theta|y)$ and $q(\theta|y)$, averaged over $y$. This is a "global" measure relevant to developing algorithms and statistical software. But sometimes the concern is how close $p$ and $q$ are for some particular observation $y = y^{\mathrm{obs}}$. "Local" diagnostics have been developed for MCMC [14, 12, 40, 17], variational inference [41, 8] and simulation-based inference [42, 22] that try to assess $q(\theta|y^{\mathrm{obs}})$ only. It would be valuable to extend our approach to address these cases. Another future direction would be to extend our approach to posterior predictive checks [11, 9, 39] that diagnose how well the statistical model $p$ fits the observed data.

# Acknowledgement

The authors thank Bruno Regaldo-Saint Blancard for his help on galaxy clustering data and simulations. The authors thank Andreas Buja, Andrew Gelman, and Bertrand Clarke for discussions.

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

# Appendices to "Discriminative Calibration"

The Appendices are organized as follows. In Section A, we provide several extra theory results for the main paper. In Section B, we discuss our experiment and implementations. In Section C, we provide proofs for the theory claims.

## A    Additional theory results

**Theorem 7** (Closed form expression of the divergence). *Consider the label generating process in Section 4. Let $\pi(t, \phi)$ be the joint density of the label and features after mapping $\Phi$, i.e., the distribution of the $(t, \phi)$ generated by the simulation process:*

1. *sample $(\theta, y) \sim p(\theta, y)$;*

2. *sample $(\tilde{\theta}_1, \ldots, \tilde{\theta}_M)$ from $q(\theta|y)$;*

3. *generate $((t_1, \phi_1), \ldots, (t_L, \phi_L))) = \Phi((y, \theta, \tilde{\theta}_1, \ldots, \tilde{\theta}_M))$;*

4. *sample an index $l$ from Uniform$(1, \ldots, L)$;*

5. *return $(t_l, \phi_l)$.*

*We define*

$$\pi_k(\phi) = \pi(\phi|t = k) = \int_y \pi(y)\pi(\phi|y, t = k)dy$$

*to be the y-**averaged law** of $\phi|t = k$. Note that $y$ has been averaged out.*

*Then classification divergence defended through (7) in Theorem 1 has a closed form expression, a Jensen–Shannon divergence in this projected $\phi$ space.*

$$D^{\text{opt}}(p, q, \Phi) = \sum_{k=0}^{K-1} w_k \text{KL} \left( \pi_k(\cdot) \;||\; \sum_{j=0}^{K-1} w_j \pi_j(\cdot) \right). \tag{13}$$

This Theorem 7 gives a connection between our divergence and the familiar Jensen–Shannon divergence, which is well known to be linked to IID sample classification error (note that the classification examples we generate are not IID for they share $y$).

As a special case, when $K = 2$, $w_0 = w_1 = 1/2$, we recover the symmetric two-group Jensen–Shannon divergence, or JSD. In general, between two densities $\pi_1(x)$ and $\pi_2(x)$, the JSD is

$$\text{Jensen Shannon divergence}(\pi_1, \pi_2) = \frac{1}{2}\text{KL}\left(\pi_1 \;||\; \frac{\pi_1 + \pi_2}{2}\right) + \frac{1}{2}\text{KL}\left(\pi_2 \;||\; \frac{\pi_1 + \pi_2}{2}\right). \tag{14}$$

As before, the standard notation [5] of conditional divergence is defined as taking expectations over conditional variables:

$$\begin{aligned}\text{JSD}(\pi_1(x|z), \pi_2(x|z)) =& \frac{1}{2}\text{KL}\left(\pi_1(x|z) \;||\; \frac{\pi_1(x|z) + \pi_2(x|z)}{2}\right) \\ &+ \frac{1}{2}\text{KL}\left(\pi_2(x|z) \;||\; \frac{\pi_1(x|z) + \pi_2(x|z)}{2}\right).\end{aligned} \tag{15}$$

**Theorem 8** (weighting). *The binary label mapping generates $M$ paris label-1 examples and one pair of label-0 examples per simulation. We can reweight the binary classifier by letting the ELPD be $\mathbb{E}[\frac{C}{M+1}\mathbb{1}(t = 1)\log p(t = 1|c(\phi)) + \frac{CM}{M+1}\mathbb{1}(t = 0)\log p(t = 0|c(\phi))]$, where $C = \frac{(M+1)^2}{2M}$ is a normalizing constant.*

*That is if we modify the training utility function or the validation data LPD to be:*

$$\frac{1}{n}\sum_{i=1}^n \left( \frac{C}{M+1}\mathbb{1}(t_i = 1)\log \Pr(t = 1|c(\phi_i)) + \frac{CM}{M+1}\mathbb{1}(t_i = 0)\log \Pr(t = 0|c(\phi_i)) \right).$$

*then the resulting binary classification divergence in Thm. 1, i.e.,*

$$\text{weighted ELPD} + \log 2,$$

*is the conditional Jensen Shannon divergence* (15).

### A.1 SBC test and generative classifier

In Example 3, we stated the intuitive connection between the SBC rank test and using rank as the feature in a "*generative*" classifier (in contrast to our "discriminate" approach). Now we make such comparison more precise: Assuming the parameter $\theta$ is one-dimensional. This is what SBC does: From $p$ we obtain $\theta, y$, and for each $\theta$ we compute the rank of it in the paired $q$ samples who shared the same $y$: $r = \sum_{m=1}^{M} \mathbb{1}(\theta \leq \tilde{\theta}_m)$. Repeating this simulation many times, we obtain a sample of $r$, and SBC will test if such $r$ is discrete-uniformly distributed. This test could be done by simulating a reference distribution that matches the null. We do so by generating the ranks in the $q$ samples: $\tilde{r}_n = \sum_{m \neq n} \mathbb{1}(\tilde{\theta}_n \leq \tilde{\theta}_m) + \mathbb{1}(\tilde{\theta}_n \leq \theta)$. To be clear, there are other ways to generate reference samples, but all it matters is that the reference distribution matches the distribution of $r$ under the null. The uniformity test in SBC now becomes to test if $r$ and $\tilde{r}$ have the same distribution. For example, we can do a two-sample Kolmogorov–Smirnov test or a permutation test.

In comparison, this is what naive Bayes would do: the naive Bayes classifier first estimates the two conditional distributions: $\Pr(r|t = 0)$ from the empirical distribution of the prior rank $r$, and $\Pr(r|t = 1)$ from the empirical distribution of $\tilde{r}$. Then to test if $\Pr(r|t = 0) = \Pr(r|t = 1)$ becomes the same to testing if the empirical distribution of $r$ is the same as the empirical distribution of $r$—the same task as in SBC. The point is that given any two-sample based test,

$$\text{SBC + sample-based test}$$

is *operationally* the same as

$$\text{only use rank + naive Bayes classifier + density estimated by histogram + sample-based test.}$$

## B Implementations

### B.1 A cheat sheet of useful pre-learned features

Table 1 summarizes how to use the statistical feature (whenever available) in both the binary and multiclass classifiers.

| | binary | multiclass |
|---|---|---|
| full feature | $(\theta, y)$ | $(\theta_1, \ldots, \theta_K, y)$ |
| MCMC | $p(\theta, y),\ r(\theta, y)$ | $p(\theta_t, y)\ r(\theta_t, y), 1 \leq t \leq K$ |
| VI | $p(\theta, y),\ q(\theta|y)$ | $p(\theta_t, y)\ q(\theta_t|y), 1 \leq t \leq K$ |
| likelihood-free | $p(\theta),\ q(\theta|y)$ | $p(\theta_t),\ q(\theta_t|y), 1 \leq t \leq K$ |

Table 1: A cheat sheet of useful pre-learned features

### B.2 Code and data

We share the python and Jax implementation of our binary and multiclass calibration code in https://github.com/yao-yl/DiscCalibration.

In the MLP training we include a standard $L^2$ weight decay (i.e., training loss function = cross entropy loss + tuning weight $* L^2$ penalization). We tune the weight of the decay term by a 5 fold cross-validation in the training set on a fixed grid $\{0.1, 0.01, 0.001, 0.0001\}$.

In the cosmology experiment in Section 5, we use one-hidden-layer MLP with 64 nodes to parameterize the classifier with the form (11), with additional pre-learned features such as $\log q(\theta|y)$ added as linear features. The approximate log density $\log q(\theta|y)$ is known (i.e., we can evaluate the for any $\theta$ and $y$, or at least least up to a constant) in either the normalizing flow or the ensemble of the normalizing flows. One classification run with roughly one million examples took roughly two hour

|  | model 1 ($d_y$=38) | model 2 ($d_y$=151) | model 3 ($d_y$=1031) |
|---|---|---|---|
| 1-flow | 0.008 (0.003) | 0.05(0.006) | 0.0001 (0.0002) |
| 5-ensemble | 0.001 (0.002) | 0.008(0.003) | 0.001( 0.001) |

Figure 8: *The estimated divergence of two simulation based inferences (either using one flow or an ensemble of five flows) in three cosmology models. These three models have the same parameter space and involve expensive simulations. Here we apply our classifier approach and estimate the Jensen–Shannon divergence (and standard deviation) using weighted binary classifiers from $S = 2000, M = 500$ simulations: that is $2000 * 501$=1 million classification examples. In this table, we run this classifier by the black-box multilayer perceptron (MLP) on the full $(\theta, y)$ vector, while the table we have in the main paper further adds $\log q(\theta|y)$ as a pre-learned feature since it is known in the normalizing flows. In this black box estimate table, we would pass a t-test and not even detect miscalculation. Adding a statistically-meaningful feature greatly improves the classifier prediction, and tightens the bound of the divergence estimate.*

cpu time on a local laptop. It would be more efficient to run the classification on GPU, thought the classification cost is overall negligible compared with the simulation step $y|\theta$ which is pre-computed and stored.

In the closed-form Gaussian experiment in Section 5, we consider the easiest setting: $\theta \in \mathbb{R}^d \sim \mathrm{MVN}(\mathbf{0}, \mathrm{Id}_d)$ and a normal data model $y|\theta \sim \mathrm{MVN}(\theta, \mathrm{Id}_d)$. Indeed, we have kept both the true distribution and the sampling corruption mean-field to make the calibration task easier for the traditional SBC rank-test. The power of the traditional SBC rank-test would be even worse if we add some interaction terms in the corrupted samples $q(\theta|y)$, while our method leans the joint space sampling quality by default. The exact posterior $p(\theta|y)$ is known but we pretend we cannot evaluate it. We set $d = 16$ to be comparable with the real data example. The right panel validates Thm. 3 fixed the simulation size $S = 5000$ and vary $M$. We have tried other fixed $M = 1000$ but it seems the resulting graph is very similiar. The confidence band in Figure 3 and Figure 4 is computed using 1.96 times standard error from repeated experiments.

Table 8 shows the naive classifier: using the bandbox MLP without the linear features. The estimate is rather loose compared with the table we obtained in Section 5 which used statistically-meaningful features.

## C   Proofs of the theorems

We prove the theorems in the following order: Thm. 7, Thm. 1, Thm. 8, Thm. 6, Thm. 3, Thm. 5, Thm. 4, and finally Thm. 2.

## Contents

### C.1   The classification divergence

To prove our Theorem 1, we first state an equivalent but a more general theorem. Note that we are allowing non-IID classification examples for the shared $z$ variables in this theory.

**Theorem 9** (The general relation between divergence and classification error). *Define a simulation process $\pi(z, k, x)$ through the following process, among the three random variables, $z \in \mathbb{R}^n$, $x \in \mathbb{R}^d$ can have any dimension, and $k \in \{1, 2, \ldots, K\}$ is an integer.*

1. *Sample $z \sim \pi(z)$*

2. *Sample $k \sim \text{Categorical}(w)$, where $w$ is a simplex.*

3. *Sample $x \sim \pi_k(x|z)$*

4. *Return $(z, k, x)$*

*Define $\text{U}(\pi)$ to be the expected log probability or cross-entropy of an optimal classifier trained to predict $k$ from $(x, z)$, i.e.*

$$\text{U}(\pi) = \max_c \mathbb{E}_\pi U(k, c(x, z)),$$

*where $U(k, c) = \log c(k)$. Then*[3]

$$\text{U}(\pi) - \sum_{k=1}^K w_k \log w_k = \sum_{n=1}^K w_k \text{KL}\left(\pi_n(x|z) \,\|\, \pi(x|z)\right)$$

*where $\pi(x|z) = \sum_{k=1}^K w_k \pi_k(x|z)$ is the mixture over all groups.*

*We define $D(\pi) = \text{U}(\pi) - \sum_{k=1}^K w_k \log w_k$. Then this $D$ has the following properties:*

1. *(lower bound). $D(\pi) \geq 0$. It achieves zero if and only if all $p_k$ are the same, i.e., $\Pr(X^k \in A) = \Pr(X^j \in A)$ for any measurable set A.*

2. *(upper bound). $D(\pi) \leq \sum_{n=1}^K w_k \log w_k$. This maximum is achieved is $p_k$ are disjoint, i.e., $\Pr(X^k \in A)\Pr(X^j \in A) = 0$ for any measurable set A.*

3. *(alternative upper bound when the difference is small). $D(\pi) \leq \max_{j,k} \text{KL}(\pi_k(x|z), \pi_j(x|z))$.*

4. *(reparametrization invariance). For any fixed bijective transformation $x \mapsto g(x)$, let $p_k^g$ be the law of $g(x^k)$, then $D(p_k^g, \ldots, p_k^g) = D(p_k, \ldots, p_k)$.*

5. *(transformation decreases divergence). For any fixed transformation $x \mapsto g(x)$, let $p_k^g$ be the law of $g(x^k)$, then $D(p_k^g, \ldots, p_k^g) \leq D(p_k, \ldots, p_k)$.*

*Proof.* The (unknown) optimal conditional classifier is the Bayes classifier:

$$\hat{\Pr}(t = k|x, z) = \frac{w_k \pi_k(x|z)}{\sum_j w_k \pi_j(x|z)}.$$

Let $r(x|z) \sum_{j=1}^K w_k \pi_j(x|z)$ be mixture density marginalized over the group index $k$.

---

[3]We emphasize that we are using the notation of conditional KL divergence:

$$\text{KL}\left(\pi_n(x|z) \,\|\, \pi(x|z)\right) := \int \left[p(z) \int \pi_k(x|z) \log \left(\frac{\pi_k(x|z)}{r(x|z)}\right) dx\right] dz.$$

The expected log predictive density of the optimal classification is then

$$
\begin{aligned}
\text{ELPD}(\hat{\Pr}(t=k|x,z)) &= \mathbb{E}_z \int r(x|z) \sum_{k=1}^{K} \left[ \frac{w_k \pi_k(x|z)}{r(x|z)} \log \left( \frac{w_k \pi_k(x|z)}{r(x|z)} \right) \right] dx \\
&= \mathbb{E}_z \sum_{k=1}^{K} w_k \int \pi_k(x|z) \log \left( \frac{\pi_k(x|z)}{r(x|z)} + \log w_k \right) dx \\
&= \mathbb{E}_z \sum_{k=1}^{K} w_k \log(w_k) + \sum_{n=1}^{K} w_k \, \mathbb{E}_z \int \pi_k(x|z) \log \left( \frac{\pi_k(x|z)}{r(x|z)} \right) dx \\
&= \sum_{k=1}^{K} w_k \log(w_k) + \sum_{n=1}^{K} w_k \text{KL} \left( \pi_k(x|z) \,,\; r(x|z) \right).
\end{aligned}
$$

When $\text{ELPD} - \sum_{k=1}^{K} w_k \log(w_k) = 0$, $\sum_{k=1}^{K} w_k \pi_k(x|z) = 0$, hence these $K$ conditional densities $\pi_k(x|z)$ are the same almost everywhere.

The reparametrization invariance is directly inherited from KL diverge.

The upper bound is the consequence of $\text{ELPD} \le 0$ for category classification and the lower bound is that KL divergence $\ge 0$. $\qquad\square$

Let's emphasize again that we are following the notation of conditional divergence in [5], such that for any two joint density $p(theta, y)$ and $q(\theta, y)$, the conditional KL divergence is $\text{KL}(p(\theta|y)\|q(\theta|y)) :=$ $\mathbb{E}_{p(y)} \mathbb{E}_{p(y|\theta)} \log \frac{p(y|\theta)}{q(y|\theta)}$, an expectation is taken over $y$.

The procedure in Theorem 9 can reflect many different generating processes. Let's recap the examples we show in Section 2.

In **Example 1** of the main paper, the binary case, $K = 2$, and we generate $M$ samples from $q$. $\pi_1 = p(\theta|y)$, $\pi_2 = q(\theta|y)$, $w_1 = 1/M$, $w_2 = (M-1)/M$. Our simulation process in example 1 is equivalent to

1. Sample $y \sim$ marginal $p(y)$,
2. Sample $k \sim \text{Categorical}(w)$, where $w_1 = 1/M$, $w_2 = (M-1)/M$.
3. Sample $x \sim \pi_k(x|z)$, where $\pi_1 = p(\theta|y)$, $\pi_2 = q(\theta|y)$.
4. Return $(z, k, x)$

As a direct consequence of Theorem 9, the resulting divergence in the binary case is

$$
D_1(p, q) = w_1 \text{KL} \left( p(\theta|y) \,\|\, r(\theta|y) \right) + w_2 \text{KL} \left( q(\theta|y) \,\|\, r(\theta|y) \right),
$$

where $r = w_1 p + w_2 q$.

In **Example 2**: Binary label with no $y$, we generate $M$ samples from $q$. $K = 2$, $\pi_1 = p(\theta)$, $\pi_2 = q(\theta)$, $w_1 = 1/(M+1)$, $w_2 = M/(M+1)$. Our simulation process in example 2 is equivalent to

1. Sample $k \sim \text{Categorical}(w)$, where $w_1 = 1/M$, $w_2 = (M-1)/M$.
2. Sample $x \sim \pi_k(x)$, where $\pi_1 = p(\theta)$, $\pi_2 = q(\theta)$.
3. Return $(k, x)$

From Theorem 9, the resulting divergence reads

$$
D_3(p, q) = w_1 \text{KL} \left( p(\theta) \,\|\, r(\theta) \right) + w_2 \text{KL} \left( q(\theta) \,\|\, r(\theta) \right)
$$

In the multivariate **Example 4**, $K = M + 1$, and the individual density is

$$
\pi_k(\theta_{1:K}) = p(\theta_k|y) \prod_{m \ne k} q(\theta_m|y)
$$

.

From Theorem 9, the resulting divergence reads

$$D_4(p,q) = \text{KL}\left(p(\theta_1|y)\prod_{m>1}q(\theta_m|y) \,\|\, \frac{1}{K}\sum_{k=1}^{K}p(\theta_k|y)\prod_{m\neq k}q(\theta_m|y)\right)$$

We give more results on this $D_4$ in Section C.4.

The discriminative calibration is designed to detect and gauge the difference between $p(\theta|y)$ and $q(\theta|y)$. From now on we use a long vector $\theta_{1:M+1}$ (the index starts by 1) to represent the simulation draw. Each *simulation draw* first generates one $\theta_1$ from the prior, one data point y from the sampling distribution $p(y|\theta_1)$, and finally $M$ draws from the approximate inference $\theta_{1,...,M+1} \sim q(\theta|y)$.

The "label generating process" $\Phi$ takes a simulated draw as its input and returns a vector of features $\phi$ and labels $t$.

$$\Phi : (y, \theta_{1,...,M+1}) \mapsto \{(\phi_1, t_1), \ldots, (\phi_K, t_K)\}.$$

The classifier will then see this feature and labels (the index $k$ is disregarded) across all simulations.

We only ask that the map $\Phi$ be within in set $\mathbb{F}$ : satisfying that:

$$\mathbb{F} : \quad \phi|y \text{ is independent of } t \text{ if } p(\theta|y) = q(\theta|y), \text{ a.e.}$$

**Proof of Theorem 1.**

*Proof.* The classifier sees features $\phi$ and labels $t$. Given a $t$, $(\phi, y)$ are IID draws from $p(y)\pi(\phi|y, t)$. In case $y$ is not observable by the classifier, $\phi|t$ are IID draws from $\int_y p(y)p(\phi|y, t)$.

To use Theorem 9, let $\mathbf{x}^k = \{(\phi_j, y)|t_j = k\}$ and $w_k = \Pr(t = k)$. The divergence from Theorem 9: reads

$$D^{\text{opt}}(p, q, \Phi) = \sum_{n=1}^{K}w_k\text{KL}\left(\pi_1(\phi), \sum_{k=1}^{K}w_k\pi_1(\phi)\right),$$

in which $\pi_k(\phi) = p(\phi|t = k) = \int_y p(y)\pi(\phi|y, t = k)dy$ is the $y$-averaged $\phi$ margin. This is essentially Theorem 7.

If $p = q$, from $\mathbb{F}$, the law of $\phi, y$ is independent of $t$ ($y$ is always independent of $t$ marginally), so that $\pi_k(\phi) = \int_y p(y)\pi(\phi|y, t = k)dy$ is also independent of $t$, as it only depends on $y$ and $\phi$, hence the divergence remains 0. $\square$

## C.2 Reweighting (Theorem 8) leads to symmetric divergence

We now prove Theorem 8. The binary label mapping generates $M$ pairs label-1 examples and one pair of label-0 examples per simulation. We can reweight the binary classifier by letting the ELPD be $\mathbb{E}[\frac{C}{M+1}\mathbb{1}(t = 1)\log p(t = 1|c(\phi)) + \frac{CM}{M+1}\mathbb{1}(t = 0)\log p(t = 0|c(\phi))]$, where $C = \frac{(M+1)^2}{2M}$ is a normalizing constant. We want to prove that after weighting, the classification divergence is symmetric Jensen shannon distance, as if we have balanced classification data.

*Proof.* After the label-reweighing, the optimal classifier is the balanced-class classifier

$$c(t = 1|\phi) = \frac{\Pr(\phi|t = 1)}{\Pr(\phi|t = 1) + \Pr(\phi|t = 0)}.$$

The re-weighted ELPD in the population is

$$\text{ELPD} = \mathbb{E}_\phi[\Pr(t = 0|\phi)\frac{CM}{M+1}\log c(t = 0|\phi) + \Pr(t = 0|\phi)\frac{C}{M+1}\log c(t = 1|\phi)]$$

$$= \mathbb{E}_y\left[\frac{p(\theta|y) + Mq(\theta|y)}{M+1}\left(\frac{p(\theta|y)}{p(\theta|y) + Mq(\theta|y)}\frac{CM}{M+1}\log\frac{p(\theta|y)}{2r(\theta|y)} + \frac{Mq(\theta|y)}{p(\theta|y) + Mq(\theta|y)}\frac{C}{M+1}\log\frac{q(\theta|y)}{2r(\theta|y)}\right)\right]$$

$$= \mathbb{E}_y\left[C\frac{2M}{(M+1)^2}\left(\frac{p(\theta|y)}{2}\log\frac{p(\theta|y)}{2r(\theta|y)} + \frac{q(\theta|y)}{2}\log\frac{q(\theta|y)}{2r(\theta|y)}\right).\right]$$

With an appropriate normalizing constant $C = \frac{(M+1)^2}{2M}$,

$$\text{ELPD} = \mathbb{E}_y \left[ \frac{1}{2} p(\theta|y) \log \frac{p(\theta|y)}{r(\theta|y)} + \frac{1}{2} q(\theta|y) \log \frac{p(\theta|y)}{r(\theta|y)} \right] - \log 2.$$

That is, the reweighted ELPD + $\log 2$ is the conditional Jensen Shannon divergence (15) between $p(\theta|y)$ and $q(\theta|y)$. $\qquad\square$

## C.3 MCMC without thinning (Theorem 6)

To diagnose MCMC, we sample $(\theta, y) \sim p(\theta, y)$, and sample (potentially autocorrelated) draws $(\tilde{\theta}_1, \ldots, \tilde{\theta}_M)$ from a MCMC sampler whose stationary distribution we believe is $q(\theta|y)$ (i.e., marginally $\tilde{\theta}_i$ is from $q(\theta|y)$), and generate examples $((t_1, \phi_1), \ldots, (t_{M+1}, \phi_{M+1}))$, from the multi-class permutation (see definition in Example 4), such that

$$\phi = (\theta_0, \theta_1, \ldots, \theta_M).$$

Then we run an exchangeable classifier parameterized by

$$\Pr(t = k|(\theta_0, \theta_1, \ldots, \theta_M, y)) = \frac{\exp(g(\theta_k, y))}{\sum_{k'=0}^{M} \exp(g(\theta_{k'}, y))}, \qquad (16)$$

where $g$ is any $\Theta \times \mathcal{Y} \to \mathbb{R}$ mapping to be learned.

Note that here $\Pr(t = k|(\theta_0, \theta_1, \ldots, \theta_M, y))$ is the classifier model we restrict it to be, not the true population $\pi$. In general $\pi(t = k|(\theta_0, \theta_1, \ldots, \theta_M, y)) = \neq \Pr(t = k|(\theta_0, \theta_1, \ldots, \theta_M, y))$ Roughly speaking, the optimal classifier should be the Bayes classifier projected to the restricted space (16), and we need to proof that this *projected restricted* solution turns out to be the same as the IID case 10, which is not trivial in general.

*Proof.* Intuitively, this separable network design (16) avoids the interaction between $\tilde{\theta}_m$ with $\tilde{\theta}_{m'}$ and disallows the network to predict $t$ based on the autocorrelation or clustering of $\tilde{\theta}$.

Because of the permutation design and we define $q$ to be the marginal distribution of $\tilde{\theta}$, first, we know that

$$\pi(\theta_k|y, t = k) = p(\theta_k|y), \quad \pi(\theta_k|y, t \neq k) = q(\theta_k|y),$$

in the MCMC population $\pi$ (there is no need to address the joint for now).

From this we obtain the conditionals in the population

$$\pi(t = k|\theta_k, y) = \frac{p(\theta_k|y)}{p(\theta_k|y) + Mq(\theta|y)}$$

and further the ratios for any $m, k$ withtin the index set $\{0, 2, \ldots, M\}$ and $m \neq k$:

$$\pi(t = k|\theta_k, \theta_m, y) = \frac{p(\theta_k|y)q(\theta_k|y)}{p(\theta_k|y)q(\theta_m|y) + q(\theta_k|y)p(\theta_k|y) + (M-1)q_{12}(\theta_k, \theta_m)}.$$

Here $q_{12}(\theta_k, \theta_m)$ is the joint distribution of two out of $M$ draws from the MCMC sampler $q(\theta|y)$, which would often not be the same as $q(\theta_k|y)q(\theta_m|y)$ because of the autocorrelation of Markov chains. We do not need to specify the form of this joint. The key blessing is that when $\theta_k$ is from $p(\theta|y)$ and $\theta_m$ is from $q(\theta|y)$, then they are independent (the numerator).

Next, from the line above we obtain the ratio estimate in the true population $\pi$:

$$\frac{\pi(t = k|\theta_k, \theta_m, y)}{\pi(t = m|\theta_k, \theta_m, y)} = \frac{p(\theta_k|y)q(\theta_k|y)}{q(\theta_k|y)p(\theta_k|y)} = \frac{p(\theta_k|y)/q(\theta_k|y)}{p(\theta_m|y)/q(\theta_m|y)}. \qquad (17)$$

Now we project the restricted classifier (16). Intuitively, the separable classifier "almost" only depends on $\theta_t$, except for the normalizing constant. We can remove the dependence on the normalizing constant by specifying the ratio

$$\frac{\Pr(t = k|(\theta_0, \theta_1, \ldots, \theta_M, y))}{\Pr(t = m|(\theta_0, \theta_1, \ldots, \theta_M, y))} = \frac{\exp(g(\theta_k, y))}{\exp(g(\theta_m, y))}.$$

Marginalizing out all other components, we obtain

$$\frac{\Pr(t = k | (\theta_k, \theta_m, y))}{\Pr(t = m | (\theta_k, \theta_m, y))} = \frac{\exp(g(\theta_k, y))}{\exp(g(\theta_m, y))}. \tag{18}$$

Matching the restriction (18) with the true population (17), the restricted projection is attainable if and only if

$$\exp(g(\theta_t, y) = p(\theta_t, y)/q(\theta_t|y),$$

so that the optimal classifier needs to be

$$\Pr(t|\theta_1, \ldots, \theta_K, y) = \frac{p(\theta_t, y)/q(\theta_t|y)}{\sum_{k=1}^{K} p(\theta_k, y)/q(\theta_k|y)}.$$

It happens that this MCMC restricted optimal classifier matches the IID optimal classifier (10). It follows from the proof in Theorem 9 that the classification divergence using the restricted classifier is still $D_4(p, q)$, as if $\{\tilde\theta_m\}_{m=1}^{M}$ are IID samples from $q(\theta|y)$. □

### C.4 Large sample limit and rate as $M \to \infty$ (Theorem 3)

In the multivariate **Example 4**, from each draw $\theta, y, \tilde\theta_{1:M}$, we generate $M + 1$ examples from $K := M + 1$ classes; one label from each. We will use the index starting from 1 in this subsection. For the $k$-th example, $t_k = k$, and the permutation for classifier reads $\phi_k$ is a vector including $y$ and $K$ copies of $\theta$, where the $k$-th copy is the prior draw, and the remaining $\theta$ are from $q(|y)$. Slightly abused the notation, in this subsection, we call this long feature vector as $\theta_{1:K}$; it is a vector in the $\Theta^{M+1}$ space. We now prove Theorem 3.

*Proof.* First, we write the true conditional label probability in this process:

$$\pi(t|\theta_{1:K}, y) = \frac{p(\theta_t|y) \prod_{j \neq t} p(\theta_j|y)}{\sum_{t'} p(\theta_{t'}|y) \prod_{j \neq t'} p(\theta_j|y)} = \frac{p(\theta_t|y)/q(\theta_t|y)}{\sum_j p(\theta_j|y)/q(\theta_j|y)}.$$

Plug it as the classifier, we obtain the optimal ELPD or negative cross entropy: ELPD = $\mathbb{E}\log\pi(t|\theta_{1:K}, y)$,

$$\begin{aligned}
\text{ELPD} &= \mathbb{E}\,\frac{p(\theta_t|y)/q(\theta_t|y)}{\sum_{k=1}^{K} p(\theta_k|y)/q(\theta_k|y)} \\
&= \mathbb{E}\log\frac{p(\theta_t|y)}{q(\theta_t|y)} - \mathbb{E}\log\sum_k p(\theta_k|y)/q(\theta_k).
\end{aligned}$$

The first term above is simply

$$\mathbb{E}\log\frac{p(\theta_t|y)}{q(\theta_t|y)} = \text{KL}\,(p(\theta|y)||q(\theta|y))$$

According to our definition (4), the divergence is ELPD offset by an entropy term,

$$D_4 := \text{ELPD} + \log K.$$

We now derive the limit of $D_4 - \text{KL}\,(p(\theta|y)||q(\theta|y))$ when $M \to \infty$ (or equivalently $K = M+1 \to \infty$)

$$\begin{aligned}
\Delta &:= D_4 - \text{KL}\,(p(\theta|y)||q(\theta|y)) \\
&= \text{ELPD} + \log K - \text{KL}\,(p(\theta|y)||q(\theta|y)) \\
&= \log K - \mathbb{E}\log\left(\sum_k \frac{p(\theta_k|y)}{q(\theta_k|y)}\right) \\
&= -\mathbb{E}\log\left(\frac{1}{K}\sum_{k=1}^{K}\frac{p(\theta_k|y)}{q(\theta_k|y)}\right)
\end{aligned}$$

Given any label value $1 \leq\leq K$, $\theta_t \sim p(\cdot|y)$, and all the remaining $\theta_j \sim q(\cdot|y)$ for $j \neq t$.

Let

$$X_k = \frac{1}{K} \sum_{k=1}^{K} \frac{p(\theta_k|y)}{q(\theta_k|y)} = \frac{1}{K} \frac{p(\theta_t|y)}{q(\theta_t|y)} + \frac{1}{K} \sum_{k \neq t} \frac{p(\theta_t|y)}{q(\theta_t|y)}.$$

The first term

$$\frac{1}{K} \sum_{k=1}^{K} \frac{p(\theta_t|y)}{q(\theta_t|y)} \to 0.$$

The second term, call it $\Delta_2$ is the mean of IID means as $\theta_j \sim q(\cdot|y)$ for $j \neq t$, the law of large number yields

$$\Delta_2 := \frac{1}{K} \sum_{k \neq t} \frac{p(\theta_t|y)}{q(\theta_t|y)}, \quad \Delta_2 \to \mathbb{E}_{x \sim q(x|y)} \frac{p(x|y)}{q(x|y)} = 1.$$

This proves that

$$\Delta \to 0, \text{ as } K \to \infty.$$

Hence,

$$D_4 - \text{KL}\left(p(\theta|y)||q(\theta|y)\right) \to 0,$$

which finished the first part of the proof.

Now let's derive the rate. Under regularity conditions, for example, the variance of density ratio is bounded, i.e., there exists a constant $C < \infty$, such that for all $y$,

$$\text{Var}_{\theta_t \sim q(\theta_t|y)} \left( \frac{p(\theta_t|y)}{q(\theta_t|y)} \right) < C,$$

then CLT holds, such that the second term above has a normal limit,

$$\sqrt{K}(\Delta_2 - 1) \to \text{normal}(0, \sigma^2), \text{ in distribution,}$$

where

$$\sigma^2 = \text{Var}_q\left(\frac{p(\theta_t|y)}{q(\theta_t|y)}\right)$$

$$= \mathbb{E}_{\theta,y \sim q(\theta,y)} \left( \frac{p(\theta|y)}{q(\theta|y)} - 1 \right)^2$$

$$= \mathbb{E}_y \mathbb{E}_{\theta \sim q(\theta|y)} \left( \frac{p(\theta|y)}{q(\theta|y)} - 1 \right)^2$$

$$= \chi^2 \left( p(\theta|y) \,||\, q(\theta|y) \right),$$

which is the definition of the conditional chi-squared divergence.

Consider a Taylor series expansion, $\log(1 + x) = x - \frac{1}{2}x^2 + o(x^2)$. Using the Delta method to express the log function and the Slutsky theorem to ignore the zero term, we get

$$K \mathbb{E} \log(\Delta_2) \to -\frac{\sigma^2}{2}.$$

Plug this in the definition of $\Delta$ we obtain the desired convergence rate,

$$D_4 = \text{KL}\left(p(\theta|y)||q(\theta|y)\right) - \frac{1}{2M}\chi^2\left(p(\theta|y) \,||\, q(\theta|y)\right) + o(1/M).$$

This proves the claims in Theorem 3. □

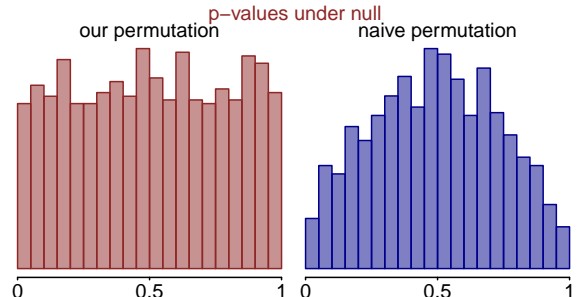

Figure 9: *Our designed permutation vs the naive permutation.*

Table 2: Batch permutation of labels

| Original batch from one run | | a batch of permuted labels | another batch of permutation |
|---|---|---|---|
| label $t$ | features $\phi$ | label $t$ | label $t$ |
| $(\theta, y)$ | 1 | 0 | 0 |
| $(\tilde{\theta}_1, y)$ | 0 | 1 | 0 |
| $(\tilde{\theta}_2, y)$ | 0 | 0 | 0 |
| $\dots$ | $\dots$ | $\dots$ | $\dots$ |
| $(\tilde{\theta}_M, y)$ | 0 | 0 | 1 |

### C.5 Valid hypothesis testing (Theorem 4)

It is not trivial to have a valid permutation test since we do not have IID examples. One naive permutation is to permutate all the labels $t$ (across batches $i$). The resulting permutation testing is not uniform under the null (Figure 9).

In contrast, when we permute the label, we only ask for within-batch permeation (permute the index $\{t_1, \dots, t_L\}$ in each batch). See Table 2 for an illustration. Let's prove this permutation is valid.

*Proof.* Under the null, $p(\theta|y) = q(\theta|y)$ almost everywhere. According to the label mapping $\Phi \in \mathbb{F}$, label $t$ is independent of $\phi$ given $y$. We first show that label $t$ is independent of $\phi$.

In general conditional independence does not lead to unconditional independence. But because here we design the generating process by $\pi(y, t, \phi) = \pi_Y(y)\pi_t(t)\pi(\phi|y, \theta)$. Under the null we have $\pi(y, t, \phi) = \pi_t(t)(\pi_Y(y)\pi(\phi|y))$. Hence $t$ needs to be independent of $\phi$.

For any classifier $c$, because $c(\phi)$ is a function of $\phi$, $c(\phi)$ and $t$ are also independent. Let $\pi_\Phi(\phi)$ be the marginal distribution of $\phi$ in $\pi$.

Now we are computing the cross entropy (with respect to the population $\pi$). LPD $= \sum_{j=1}^N U(c(\phi_n), t_n)$ where $U$ is the log score $U(P, x) = \log P(x)$. It is a random variable because we have a finite validation-set. Now we are conducting a permutation of label $t$, the permuted label $\tilde{t}$ is an independent draw from the same marginal distribution of $t$, $\pi_t(t)$. Because of the independence, $\mathbb{E}_{\phi, t} U(c(\phi), t) = \mathbb{E}_t \mathbb{E}_\phi U(c(\phi), t)$, hence the permuted $\text{LPD}_b \overset{d}{=} \text{LPD}$, where $\text{LPD}_b$ is the computed LPD in the $b$-th permutation. Therefore $\Pr(\text{LPD} \leq x) = \Pr(\text{LPD}_b \leq x)$. In expectation (with an infinitely amount of repeated random permutation), this $p$-value will be uniform on [0,1].

In practice, the computed p-value from finite permutation can only take values on a finite set $\{0, 1/B, \dots, (B-1)/B, 1\}$. More precisely, under the null hypothesis, this permutation test p-value is *discreetly-uniform* on this set. This is because for any $0 \leq m \leq B$

$$\Pr(p = m/B) = \int_0^1 \binom{B}{m} p^m (1-p)^{B-m} dp = 1/(B+1), \ \ \forall m.$$

Hence, the permutation $p$-value based on $B$ random permutations is uniformly distributed on the set $\{0, 1/B, \dots, (B-1)/B, 1\}$. □

## C.6  Why does classifier help test? (Theorem 5)

Let's recap the binary label-generating process:

1. Sample $y \sim$ marginal $p(y)$,
2. Sample $k \sim$ Bernoulli$(w)$, where $w_1 = 1/M$, $w_2 = (M-1)/M$.
3. Sample $\theta \sim \pi_k(\theta|z)$, where $\pi_0 = p(\theta|y)$, $\pi_1 = q(\theta|y)$.
4. Return $(z, k, x)$

Theorem 5 state that the optimal classifier has a sufficiency propriety in that (a) let $\hat{c}$ be the probability of label 1 in the optimal classifier as per (7), and let $\pi_c^p$ and $\pi_c^q$ be the one-dimensional distribution of this $\hat{c}(\phi)$ when $(\theta, y)$ is sampled from $p(\theta, y)$ or from $p(y)q(\theta|y)$ respectively, then (i) Conditional on the summary statistic $\hat{c}$, the label $t$ is independent of all features $\phi = (\theta, y)$. (ii) There is no loss of information in divergence as the joint divergence is the same as the projected divergence, $D_1(p, q) = D_1(\pi_c^p, \pi_c^q)$.

*Proof.* There are three random variables, $\theta$, $y$, and $t$. The optimal classifier $\hat{c}$ is the probability in this true joint:

$$\hat{c}(\theta, y) = \Pr(t = 1|(\theta, y)).$$

To show sufficiency or conditional independence, all we need to show is that, conditional on any given value of $\hat{c}$, $\Pr(t = 1|(\theta, y), c)$ does not depend on $(\theta, y)$ (in other words, $\Pr(t = 1|(\theta, y), c)$ is a function of $c$ only). This becomes obvious as

$$\Pr(t = 1|(\theta, y), c(\theta, y)) = \Pr(t = 1|(\theta, y)) = \hat{c}(\theta, y).$$

Now we prove that there is "no loss" in divergence.

$$D_1(p, q) = w \, \mathrm{KL}(p(\theta, y)||r(\theta, y)) + (1 - w) \, \mathrm{KL}(q(\theta, y)||r(\theta, y))$$

We express the first term in $\hat{c}$

$$\mathrm{KL}(p(\theta, y) \,||\, wp(\theta, y) + (1 - w)q(\theta, y))$$
$$= \mathrm{KL}(\pi(\theta, y|t = 0) \,||\, \pi(\theta, y))$$
$$= \mathrm{KL}(\pi(\hat{c}|t = 0) \,||\, \pi(\hat{c})) - \mathrm{KL}(\pi(\theta, y|\hat{c}, t = 0) \,||\, \pi(\theta, y|\hat{c}))$$

This steps uses the chain rule of KL divergence: $\mathrm{KL}[p(x, y) \,|\, q(x, y)] = \mathrm{KL}[p(x) \,|\, q(x)] + \mathrm{KL}[p(y \,|\, x) \,|\, q(y \,|\, x)]$

Using conditional independence:

$$\pi(\theta, y|\hat{c}, t = 0) = \pi(\theta, y|\hat{c}, t = 1)$$

Hence $\mathrm{KL}(\pi(\theta, y|\hat{c}, t = 0) \,||\, \pi(\theta, y|\hat{c})) = 0$. Therefore,

$$\mathrm{KL}(p(\theta, y)||r(\theta, y)) = \mathrm{KL}(\pi(\hat{c}|t = 0) \,||\, \pi(\hat{c}))$$

where $\pi(c) = \pi(\hat{c}|t = 0) + (1 - w)\pi(\hat{c}|t = 0)$

Similarly,

$$\mathrm{KL}(q(\theta, y)||r(\theta, y)) = \mathrm{KL}(\pi(\hat{c}|t = 1) \,||\, \pi(\hat{c}))$$

This proves $D_1(p, q) = D_1(\pi_c^p, \pi_c^q)$.

$\square$

## C.7  The maximum discriminative generator (Theorem 2)

We save the proof of Theorem 2 in the end for its length.

The generator $\phi$ contains a few degrees of freedom: the number of classes $K$, the number of examples $L$, and how to design and the label-feature pairs. In the binary labeling: $t_1 = 1, \phi_1 = (\theta_1, y)$ and the remaining $t_k = 0, \phi_k = (\theta_k, y)$ for $2 \leq k \leq M + 1$. The multi-class $\Phi^*$ assigns labels 1:$K$ as

$$\Phi^* : t_k = k, \phi_k = (\mathrm{Perm}_{1 \to k}(\theta_1, \dots, \theta_M), y), 1 \leq k \leq K = M + 1. \tag{19}$$

Before the main proof that the multiclass permutation creates the largest divergence, let's first convince that the multi-class classification produced higher divergence than the binary one.

In the binary classification with $M = 1$ (one draw from $p$ and one draw from $q$)

$$D_1(p, q) = \frac{1}{2} \operatorname{KL}\left(p(\theta|y), \frac{p(\theta|y) + q(\theta|y)}{2}\right) + \frac{1}{2} \operatorname{KL}\left(q(\theta|y), \frac{p(\theta|y) + q(\theta|y)}{2}\right).$$

In the multi-class classification with $M = 1$,

$$D_4(p, q) = \operatorname{KL}\left(p(\theta^1|y)q(\theta^2|y), \frac{p(\theta^1|y)q(\theta^2|y) + q(\theta^1|y)p(\theta^2|y)}{2}\right).$$

Use the joint KL > marginal KL, we have

$$D_4(p, q) \geq \operatorname{KL}\left(p(\theta^1|y), \frac{p(\theta^1|y) + q(\theta^1|y)}{2}\right).$$

Likewise,

$$D_4(p, q) \geq \operatorname{KL}\left(q(\theta^2|y), \frac{p(\theta^2|y) + q(\theta^2|y)}{2}\right).$$

Add these two lines we obtain

$$D_4(p, q) \geq D_1(p, q).$$

To prove that this multi-class permutation produced the uniformly largest divergence (across $M$, $K$, $p$, $q$) optimal, we organize the proof into lemmas 5 to 9. For notation brevity, we denote $\hat{M} := M + 1$ in these lemmas to avoid using index $M + 1$.

**Lemma 10.** *For an arbitrary integer L, any given output space $\mathcal{Y}$, and any input space $\mathcal{X}$ that has at least L elements, if there are two functions mapping $\mathcal{X}^L$ to $\mathcal{Y}$*

$$f_1, f_2 : (x_1, \ldots, x_L) \mapsto y \in \mathcal{Y}.$$

*satisfying the following propriety:*

- *for any probability distribution $\pi$ on $\mathcal{X}$, when $x_1, \ldots, x_n$ are L iid random variables with law $\pi$, $f_1(x_1, \ldots, x_L)$ has the same distribution as $f_2(x_1, \ldots, x_L)$,*

*then there must exist a permutation of $1 : L$, denoted by $\sigma(1 : L)$, such that*

$$f_2(x_1, \ldots, x_L) = f_1(x_{\sigma(1)}, \ldots, x_{\sigma(L)}).$$

*Proof.* For any $L$ distinct values $a_1, \ldots, a_L$, $a_i \in \mathcal{X}$, let $\pi$ be a mixture distribution of $L$ delta functions:

$$\pi = \sum_{m=1}^{L} \delta(a_m) p_m,$$

where the $m$-th mixture probability is

$$p_m = C \left(\frac{1}{2}\right)^{L^{m-1}}, \quad C^{-1} = \sum_{m=1}^{L} \left(\frac{1}{2}\right)^{L^{m-1}}.$$

$C$ is chosen such that $\sum_{m=1}^{L} p_m = 1$.

Now that $x_1, \ldots, x_L$ are $L$ IID random variables from this $\pi$, $f_1(x_1, \ldots, x_L)$ is also a mixture of delta functions, For any sequence of input indices $(u_1, \ldots, u_L) \in \{1 : L\}^L$,

$$\Pr(f_1(x_1, \ldots, x_L) = f_1(a_{u_1}, \ldots, a_{u_L})) = \prod_{m=1}^{L} ((C/2^{L^{m-1}})^{\sum_{j=1}^{L} 1(u_j = m)})$$

$$= C^L (\frac{1}{2})^{\sum_{m=1}^{L} (\sum_{j=1}^{L} 1(u_j = m) L^{m-1})}, \qquad (20)$$

in which the power index can be written as

$$\left(\sum_{j=1}^{L} 1(u_j = 1), \ldots, \sum_{j=1}^{L} 1(u_j = L)\right)_L := \sum_{m=1}^{L}\left(\sum_{j=1}^{L} 1(u_j = m)L^{m-1}\right)$$

as an $L$-decimal-integer.

Next, we study the law of $f_2(x_1, \ldots, , x_L)$:

$$\Pr(f_2(x_1, \ldots, , x_L) = f_2(a_1, \ldots, , a_L)) = C^L(\frac{1}{2})^{(1,1,\ldots,1)_L}.$$

Because $f_2(x_1, \ldots, , x_L)$ and $f_1(x_1, \ldots, , x_L)$ have the same distribution, $f_2(a_1, \ldots, , a_L))$ needs to match the value at which $f_1(x_1, \ldots, , x_L)$ has probability $C^L(\frac{1}{2})^{(1,1,\ldots,1)_L}$ to attain. Comparing with (20), this probability is only attained when $\sum_{j=1}^{L} 1(u_j = m) = 1, \forall m$. That is, there exists a $\sigma$, a permutation of $1:L$, such that $u_1, \ldots, u_L = \sigma(1, 2, \ldots, L)$.

Matching the value of $f_2(x_1, \ldots, , x_L)$ we obtain

$$f_2(a_1, \ldots, a_L) = f_1(a_{\sigma(1)}, \ldots, a_{\sigma(L)}).$$

Because the choice of vector $(a_1, \ldots, a_L)$ is arbitrary, we have

$$f_2(x_1, \ldots, x_L) = f_1(x_{\sigma(1)}, \ldots, x_{\sigma(L)}).$$

$\square$

Because augmentation increases divergence, for the purpose of finding the largest divergence, to produce the largest divergence, we only need to consider an augmented generator that includes all $y$

$$\Phi^{aug} : (y, \theta_{1,\ldots,L}) \mapsto \{((\phi_1, y), t_1), \ldots, ((\phi_K, y), t_K)\}.$$

It is enough to consider the generator of which $\phi_k = \phi_k(\theta_1, \ldots, \theta_L)$ are $K$ functions of $(\theta_1, \ldots, \theta_L)$.

**Lemma 11.** *For any augmented generator $\Phi^{arg}$ satisfies $\mathbb{F}$, the null, there must exists $K$ permutations $\sigma_1(1:L), \ldots, \sigma_K(1:(L))$, with the convention $\sigma_1(1:L) = 1:L$, such that*

$$\phi_k(\theta_1, \ldots, \theta_L) = \phi_1(\theta_{\sigma_k(1:(L))}).$$

*Proof.* Use Lemma (10) for $(K - 1)$ times. $\square$

**Lemma 12.** *For any augmented generator $\Phi^{arg}$ satisfies $\mathbb{F}$, all feature-label generator can be replaced by a permutation, $\phi_k(\theta_1, \ldots \theta_L) = (\theta_{\sigma_k(1)}, \ldots \theta_{\sigma_k(L)})$, while the divergence does not decrease.*

*Proof.* From the previous lemma,

$$\phi_k(\theta_1, \ldots, \theta_L) = \phi_1(\theta_{\sigma_k(1:L)}).$$

The augmented feature is now $(\phi_1(\theta_{\sigma_k(1:L)}), y)$, a transformation of $(\theta_{\sigma_k(1:L)}), y)$. Using the raw feature $(\theta_{\sigma_k(1:L)}), y)$ keeps the divergence non-decreasing. $\square$

Now that we only want to consider permutation-based generators: there exists a $K$, and $K$ permutations $\sigma_k(1:L)$.

$$\Phi^{aug} : (y, \theta_{1,\ldots,L}) \mapsto \{((\phi_1, y), t_1), \ldots, ((\phi_k, y), t_k)\}.$$

$$\phi_k(\theta_1, \ldots, \theta_L) = \theta_{\sigma_k(1:L)}.$$

Given a $p \neq q$, any permutations $\theta_{\sigma_i(1:L)}$ contains one copy from $p$ and $(L - 1)$ copies from $q$. It suffices to only consider those permutation $\theta_{\sigma_i(1:L)}$ whose distributions are unique.

**Lemma 13.** *Given any $p \neq q$ and $L$ fixed, assuming all $\theta_{\sigma_i(1:L)}$ has different distributions, then $D(q, p, \Phi^{arg})$ is an increasing function on $K$.*

*Proof.* Using that joint KL is always not smaller than the KL divergence of sub-coordinates. □

**Lemma 14.** *Among all permutations* $\sigma(\theta_1, \ldots \theta_L)$*, the maximum number of distinct distributions are* $K = L$.

*Proof.* The total number of permutations is $L!$. Because $\theta_{2:L}$ are IID given $y$, the permutation of index $2:L$ does not change the distribution. When $p \neq q$, the total number of distributionally distinct permutations is $L!/(L-1)! = L = M + 1$. □

It is clear that the proposed multi-class permutation $\Phi^*$ attains this maximum number of distinct distributions, which proves its optimality among all generators from all previous lemmas, thereby the proof of our Theorem 2.

