# OpenReview forum: "Discriminative Calibration: Check Bayesian Computation from Simulations and Flexible Classifier"
_NeurIPS.cc/2023/Conference — NeurIPS 2023 poster_

### Official Review · Reviewer_aBBC · 2023-07-06

**Soundness:** 4 excellent
**Presentation:** 4 excellent
**Contribution:** 3 good
**Rating:** 7
**Confidence:** 3

**Summary:**

This paper presents a classifier based approach to provide a measure of miscalibration in Bayesian computation, including for methods such as Approximate Bayesian Computation (ABC) and Simulation-Based Inference (SBI) methods like neural posterior estimation. The method enables the test statistic to be learned from data and  provides an interpretable divergence measure. The method is a form of two sample testing applied in the amortized and simulation based inference settings. Beyond standard approaches to using classifiers for two sample testing, the authors develop several classifier based approaches which use a form a "label mapping", and provide theoretical work on the validity and efficacy of these approaches. Experiments are provided to test the method for posterior inference in cases where the posterior is known, and provide empirical verification of the theoretical results. The paper also compares to the commonly used simulation based calibration method, and show clear improvements with the methods proposed in the paper. Experiments on cosmological data are also explored.

**Strengths:**

This is a good paper, it provides:
-  a clear description of the challenge they hope to address, specifically providing a better and more statistically interpretable measure of miscalibration over simulation-based calibration
- a clear description of the proposed methods, including how to use label mapping to develop calibration based diagnostics, and how this differs from other approaches
- theoretical grounding for the calibration based measures, including the expected behaviors in large sample limits
- useful discussions on implementation details
- useful discussion on the legitimacy and power of the tests.

**Weaknesses:**

While the idea of using classifiers for two sample testing is not new, the authors do develop new methods based on label mapping. While the new classification approaches are discussed and will likely be quite useful for simulation based inference, some attempts at using classifiers for testing posterior inference quality have been performed before (albeit not directly with some of the new classification methods discussed in the work). For instance, Vandegar et. al "Neural Empirical Bayes: Source Distribution Estimation and its Applications to Simulation-Based Inference" AISTATS 2021, use the AUC of a classifier as a diagnostic in a simulation based setting.

Further development of the experiments would be quite useful. A very simple experiment is shown, allowing an exploration of the methods, but some experiments more clearly showing the quality of the method in a controlled but more complex simulation based inference setting could provide useful insights to the readers. For instance, a common example is the Simple Likelihood, Complex Posterior (SLCP) problem from Papamakarios et. al, "Sequential neural likelihood: Fast likelihood-free inference with autoregressive flows", AISTATS 2019. As it stands now, it is difficult to asses the quality of the method in the more complex cosmological data set experiment.



**Questions:**

Could you provide more details on the posterior model in the multi-variate Gaussian example?

Could you provide experiments on more complex but controlled examples, for instance the SLCP problem from Papamakarios et. al? This could provide more evidence on the utility of the method in SBI settings.

How do these methods scale with feature and parameter dimension? Does this affect the ability to attain a useful diagnostic?

You mention that the presence of nuisance parameters do not affect the quality of the diagnostics. However, the Neyman-Pearson lemma does not guarantee that the likelihood ratio is the uniform most powerful test for a given size in the presence of nuisance parameters. Perhaps I am not fully understanding the text, could you explain if this has any ramifications for the proposed tests?

Could you add references to other simulation based inference work using classifiers for diagnostics? In addition, recent work on using neural networks for two sample testing (e.g. Grosso et. al, https://arxiv.org/abs/2305.14137) may be relevant to discuss in the related work.

**Limitations:**

The authors provide clear discussions of limitations. It would be interesting to also know if there are limitations dependent on the dimension of the features or parameters, and how this may impact the number of samples needed to attain a useful test.

---

> ### Author Rebuttal · Authors · 2023-08-09
>
> We are grateful for your careful reading and constructive reviews. Please find below a detailed point-by-point response to all of your comments.
>
> **Further development of the experiments would be quite useful/more complex but controlled examples, for instance the SLCP problem?**
>
> Thank you for your suggestions on the experiments! We find the SLCP example interesting. To showcase our method on a wider range of problems, we now add three additional simulation data examples from the `sbi-benchmark` repo: the Gaussian linear, Gaussian mixture, and the SLCP example.  We run our calibration on these three datasets with varying inferences settings and we obtain positive divergence estimates when the inference is not exact. Please see our **Shared Response #1** and attached pdf file.
>
> **Could you provide more details on the posterior model in the multivariate Gaussian example?**
>
> We would like to refer to our Supplement B.2 for experiment details---It was an oversight that we did not state more clearly that this information was there and we apologize. We will state this clearly in the revision.
>
> In the closed-form Gaussian experiment in Section 5, we consider an easy example: a normal likelihood  $y\sim$MVN($\theta$, Id) and a normal prior $\theta \sim$MVN(0, Id). The true posterior is a closed-form Gaussian.  We consider a sequence of corrupted inference by adding a bias shift to the posterior mean or multiplying the posterior covariance matrix by a bias factor. Notably, we have kept this true posterior distribution and the sampling corruption mean-field to make the calibration task easier for the traditional SBC rank-test.
>
> **How do these methods scale with feature and parameter dimension? Does this affect the ability to attain a useful diagnostic?**
>
> First, traditional rank-based SBC faces challenges in high dimensions due to (1) the difficulty of checking interactions between dimensions and (2) the expense of multiple testing. Running SBC in high-dimensional problems is one of our initial motivations. In the real data cosmology example, the input dimension ($\theta \times y$ joint space) of the classifier input we trained is ~1000.
>
> Strictly speaking, high dimensions in parameters do not pose any conceptual difficulty for our framework. However, to get useful calibration it is necessary that the classifier is able to distinguish the two or multiple classes—if the classification problem is too challenging, the learned classifier could be far from the optimal one, which would mean our divergence estimate is a weaker bound and more false negatives would occur in testing. Our Section 4 offered practical recommendations to help dimension scaling, including using (1) pre-learned features for dimension reduction, (2) statistical features such as log p and log q whenever available, and (3) network symmetry.
>
> **the presence of nuisance parameters do not affect the quality of the diagnostics. However, the NP lemma does not guarantee that the likelihood ratio is the uniform most powerful test for a given size in the presence of nuisance parameters.**
>
> Thank you for raising this interesting question! In the paper, we discussed the nuisance parameter when the full parameter space can be partitioned into two parts, say $\theta$, the parameters of interest, and $\phi$, the nuisance parameters.  We are only interested in the marginal sampling quality of the $\theta$, that is if $p(\theta|y)= q(\theta|y)$.  We argue that we only need to restrict the classifier to use the target dimensions $\theta$ and $y$, while other nuisance parameters can be discarded from the classifier. As a consequence of our Theorem 1, the estimated divergence from this classifier is the conditional divergence between $p(\theta|y)$ and $q(\theta|y)$. The nuisance parameter has no ramifications.
>
> The Neyman-Pearson lemma does not apply to nuisance parameter problems as the null and alternative are composite. Here we do not have a composite hypothesis since we just look at marginal posterior distribution $\theta|y$. To be clear, when looking at the $\theta$ margin, the density ratio to be learned from the classifier is $p(\theta|y)/q(\theta|y)$, not the original full density ratio $p(\theta, \phi|y)/q(\theta, \phi|y)$, which is what the likelihood ratio means in the NP lemma.
>
> **Could you add references to other simulation based inference work using classifiers for diagnostics?  Recent work on using neural networks for two sample testing may be relevant to discuss in the related work.**
>
> Thank you for your suggestions on references, which we will add.
>
> First, using the classifier two-sample test (C2ST) is not a new idea. However, the traditional classifier two sample test does not directly apply to the SBI problem because of the sequential sampling and the autocorrelation. Our present not only formulates the SBC into a classifier-two-sample-test task and interprets the divergence in posterior distribution, but also develops the traditional C2ST to incorporate autocorrelation and extra log density information, as well as a general label mapping framework. Please see our **Shared Response #2**  for details of our in relation to C2ST.
>
> It was an oversight that we did not cite Vandegar et. al (2021) who used ROC AUC to compare (1) samples from $p(\theta|y)$ and $q(\theta|y)$, and (2) the data y and the simulated posterior predictive $\int p(y|\theta) q(\theta|y) d\theta$. We apologize for the oversight we will add in the revision. However,  Vandegar et. al (2021) addressed a different task than ours. Assessment (1) requires samples from the exact posterior $p(\theta|y)$, which is different from our intended application in diagnostics. Assessment (2) is related to the posterior predictive check (e.g., Gelman 1996),  a way to examine the model specification $p(y|\theta)$ rather than (purely) computation accuracy. Our diagnostics are only intended to diagnose the computation, not to check the model correctness (line 377).

---

> > ### Comment · Reviewer_aBBC · 2023-08-15
> > **Response**
> >
> > Dear Authors, thank you for your detailed responses. I continue to consider this a good paper and believe it should be accepted.
> >
> > In terms of nuisance parameters, indeed NP is for simple hypotheses, but my point was that only likelihood ratios with simple hypotheses will result in uniformly most powerful tests. This is as opposed to Wilk's test / likelihood ratios test that deals with nuisance parameters through maximization, but is not guaranteed to be uniformly most powerful. For marginal likelihoods, it is not clear this will be a uniformly most powerful test. Nonetheless, we may be discussing an issue that is not the most relevant point for your work.

---

> > > ### Author Response · Authors · 2023-08-15
> > >
> > > Thank you for your careful reading and insights! Let us clarify that the exact NP lemma and the uniformly most powerful test are only precisely applicable when we know the exact likelihood ratios, while in the classifier two-sample test, the density ratios are learned from finite samples. For a two-sample test, we agree that nuisance parameters can be informative too: for example, if there is a known correlation between target parameters and nuisance parameters, then ignoring the nuisance parameters may reduce the power of the test. In general, the theory of Neyman structure and the unbiased similar test generalizes the NP lemma to the existence of nuisance parameters, but these theories are beyond the scope of this paper.

---

### Official Review · Reviewer_gi2H · 2023-07-06

**Soundness:** 4 excellent
**Presentation:** 3 good
**Contribution:** 3 good
**Rating:** 7
**Confidence:** 3

**Summary:**

In this paper the authors focus on the challenge of comparing two conditional distributions p, q from their samples. In particular, this is useful as a check for bayesian computations.
To achieve this, the authors propose the use of a probabilistic classification approach where they create a new dataset combining samples from p and q and labels related to the particular distribution. Here, the authors propose different approaches and make connections to previous work.
Given the combined dataset, a classifier tries to predict the label from the features. Failing to do so, indicates that the classifier could not discriminate between the two distributions.
The performance of the probabilistic classifier is then used to estimate the divergence between the distributions and for hypothesis testing.
In the empirical section, the authors compare against SBC and show improvements in power given a reduced number of samples.

**Strengths:**

The paper analyses different approaches for the generation of samples to be used by discriminators. They then present theoretical developments and demonstrate their results empirically. The sampling approaches and theoretical treatment are novel, of high quality and well written.
The main advantage here is that even for a sub-optimal classifier, one gets improvements in data efficiency.

**Weaknesses:**

The main weakness of the paper is that in the empirical evaluation, the authors do not explore the case where the samples present more challenging behaviors such as auto-correlations, different types of imbalances, etc.

Other minor suggestions:
- The colors of the thetas in the introduction are hard to see.
- Describing why the bounds are tight could further help the reader in section 2.

**Questions:**

I'm thinking of your paper as an approach to test whether the labels and the generated features are connected. This reminds me of [1] where they evaluate the mutual information via neural networks. I'm wondering both if it makes sense in your setting to estimate the divergence via the Donsker-Varadhan representation, and in their setting if your feature generation approach is beneficial. Do you have any opinions on that?

The same has applications for other approaches on independency testing via classifiers where your feature generation approach also has potential.

[1] Belghazi, Mohamed Ishmael, et al. "Mutual information neural estimation." International conference on machine learning. PMLR, 2018.

**Limitations:**

There are no potential negative societal impacts from this work. The authors mention limitations in a future work section.

---

> ### Author Rebuttal · Authors · 2023-08-09
>
>
> We are grateful for your careful reading of our manuscript and your constructive review comments. We have now addressed your concerns to improve and clarify the manuscript. Please find below a detailed point-by-point response to all of your comments and questions.
>
> **The main weakness of the paper is that in the empirical evaluation.**
>
> Thank you for your suggestions on the experiments. To showcase our method, we now add three simulation data examples from this sbi-benchmark repo, the Gaussian linear, the Gaussian mixture, and the simple likelihood complex posterior (SLCP) example. We run our calibration method on these three datasets with varying inferences settings and it picks up positive divergence estimates when the inference is not exact. Please see our Shared response #1 and attached pdf file.
>
> **Describing why the bounds are tight could further help the reader in section 2.**
>
> Thank you for your suggestion. We use Section 2 to present four examples to help the reader quickly see the heuristics and intuition: why we can formulate the simulation based calibration into a joint space classification, and various label mapping could lead to different divergence estimates. We state the rigorous theory in Section 3, but we felt this was too abstract without having some examples first. We provide detailed derivation of all theorems including the tightness of the bound in Supplement C.
>
> **I'm thinking of your paper as an approach to test whether the labels and the generated features are connected. This reminds me of [1] where they evaluate the mutual information via neural networks. I'm wondering both if it makes sense in your setting to estimate the divergence via the Donsker-Varadhan representation, and in their setting if your feature generation approach is beneficial.**
>
>
> Thank you for your insightful remark. It is an interesting connection, which we did not anticipate! Nguyen et al. (2009) developed an M-estimation to compute the f-divergence from two samples using conjugate dual function theory. Belghazi et al. (2018) developed another tighter sample-based estimate of the two-sample KL-divergence using the Donsker-Varadhan representation. Both estimates have been applied in GAN to replace the traditional binary classifier.
>
> First, let us clarify what makes the simulation based calibration (SBC) setting different from the generic two-sample divergence computation—The SBI joint draws $(\theta, y)$ and $(\tilde \theta_m, y)$ are not IID draws from two simulators since all the $\theta$ and $\tilde \theta_m$ are paired with an identical $y$. Furthermore, when the inference $q(\theta|y)$ is MCMC, there is additional auto-correlation in $\tilde \theta_m$. Both the across-class and within-class dependence violates the IID sampling assumption in Nguyen et al. (2009) and Belghazi et al. (2018). In comparison, our proposed multiclass classification (example 4) solves both issues. Even better, our multiclass classification is always balanced. Indeed, the main motivation of our multiclass classification development is to address imbalanced and auto-correlated samples, not to purely develop a sample based KL divergence estimate, and any finite M produces a divergence metric as well.
>
> Second, similar to how we can see our method as a generalization of the C2ST, it is likely that we could adapt Nguyen et al. (2009) or Belghazi et al (2018)'s KL estimate to the SBI setting. Indeed, it is the contribution of our paper to formulate the traditionally rank- and histogram-based SBC problem into a sample-based joint-space discriminative task. Our framework is flexible to incorporate the inclusion of other sample-based divergence estimates and loss functions, such as the aforementioned sample KL estimates, Wasserstein distances, and the integral probability metrics. When some part of density information p or q is known such as in SBI and GAN, it is straightforward to include them as features in the learning of the Donsker-Varadhan representation as we have done in Section 4. We leave this extension for  future work.
>
> Lastly, the “byproduct” of our calibration method is a novel sample based KL divergence estimate using multi-class classification (example 4), which can be used in other applications. A direct consequence of our Theorem 3 is that, in a generic two-simulator setting, the divergence estimate obtained from any multi-class classifier and any simulation size M is always a lower bound to KL divergence from p to q—and this estimate becomes tight when the classifier is optimal and M goes to infinity. In the paper we have proven its convergence, yielded the convergence rate (theorem 3, where the constant is further the chi-squared divergence), and shown the empirical evidence. Perhaps the most interesting direction for future investigation (although unrelated to this paper) is to compare this new multi-class-classifier estimate to its orthogonal counterpart, the Donsker-Varadhan representation and the f-dual, in the IID sampling case (which is NOT SBC, but includes the GAN, two-sample test, and independence test) such that all three methods are applicable.

---

> > ### Comment · Reviewer_gi2H · 2023-08-14
> >
> > I thank the authors for the provided examples. I still consider the paper too be good. I will keep my score and follow the discussion, suggesting it be accepted.

---

### Official Review · Reviewer_TMBR · 2023-07-07

**Soundness:** 3 good
**Presentation:** 3 good
**Contribution:** 3 good
**Rating:** 6
**Confidence:** 3

**Summary:**

In this paper, the authors consider conducting frequentists tests for simulation-based inference. Specifically, the authors want to test if an inference engine q(theta | y) produces samples from the true posterior p(theta | y). The key idea is to use classifiers to determine similarities between samples generated from the true and approximated posterior (similar to GAN).

The foundation of the algorithm is laid in section 3, where the authors show the negative cross-entropy is associated with a generalised divergence between p and q. In particular, D4 becomes KL divergence between p and q as the number of posterior draws goes to infinity. Empirical study validate the theoretical results and shows the effectiveness of the algorithm.



**Strengths:**

1. This paper is generally well-written and the idea is intuitive and sound.
2. This paper addresses an understudied problem (SBC) in the literature, and the expansion from a single-dimensional rank statistic to a multidimensional test statistic seems to be a significant improvement
3. The algorithm provides theoretical support (Section 3) to the proposed algorithm.

**Weaknesses:**

1. From the methodological point of view, using a classifier to compare distributions is not a new idea (as the authors have discussed in Section 6.
2. When training a classifier, it seems authors are facing very imbalanced classes and training an imbalanced classifier itself can be difficult. I am wondering if this would cause issues in the tests performed later.

**Questions:**

If one wants a divergence KL(p(theta|y, q(theta|y)), one can directly approximate it without using a classifier, right?

KL(p(theta|y), q(theta|y) = KL(p(theta, y), q(theta|y)p(y)).

Suppose you can sample from p(theta, y), q(theta|y) and p(y). Then we can directly approximate above KL using f-GAN,  and perform a permutation test or compute CI using bootstrap.

This approach sounds like a simpler way to obtain a divergence measure between p and q, so I wonder why the authors did not explore this idea nor use it as a benchmark method.

----------

Revision after author's response.

This approximation cannot be easily applied due to the IID-ness and the sequential nature of SBI as authors mentioned.


**Limitations:**

Yes

---

> ### Author Rebuttal · Authors · 2023-08-09
>
>
> We are grateful for your careful reading and insightful comments. We have now addressed your concerns to improve and clarify the manuscript. Please find below a detailed point-by-point response to all of your comments and questions. We wish our explanations to help bring clarification.
>
> **From the methodological point of view, using a classifier to compare distributions is not a new idea (as the authors have discussed in Section 6.**
>
> Using the classifier two sample test (C2ST) to perform two sample test is not new. However, the traditional classifier two sample test does not directly apply to the SBI problem because of the sequential sampling and the autocorrelation. Our results not only formulate the SBC into a classifier-two-sample-test task and interprets the divergence in posterior distribution, but also develop the result to incorporate autocorrelation and extra log density information. We also develop a more general label mapping framework. This framework and the novel multiple-class classifiers approach may be applicable to other two sample test problems.
>
> Please see our **Shared Response #2**  for details on why our paper develops the classifier two sample test approach.
>
>
> **When training a classifier, it seems authors are facing very imbalanced classes and training an imbalanced classifier itself can be difficult. I am wondering if this would cause issues in the tests performed later.**
>
> We apologize for this confusion, but our novelly developed multiclass label-mapping and corresponding multiclass classifiers are always *balanced* (Example 4 in Section 2, where we create M+1 labels from M+1 classes from each simulation draw; one label from each class). Hence increasing the number of inference draws (M) is not a difficulty for us. We have developed asymptotic theory, convergence rate ($M\to \infty$) and empirical validation to this big M limit.
>
> In comparison, the naive one-to-M binary classification could run into the imbalanced classification challenges you have raised, underscoring the benefit of our generalization of label mapping. That said, our method still supports binary classifiers with reweighting (Theorem 8 in Appendix). In our experiment, we find that even under highly imbalanced binary labeling (M=1000), with appropriate reweighting, the binary classifier is still able to detect the inference flaws and output accurate divergence estimates.
>
> We would like to refer to our **Share Response #3** for a more detailed discussion on why our multiclass method leads to balanced classification.
>
>
> **[f-GAN]  If one wants a divergence KL(p(theta|y, q(theta|y)), one can directly approximate it without using a classifier, right?
> Suppose you can sample from p(theta, y), q(theta|y) and p(y). Then we can directly approximate above KL using f-GAN, and perform a permutation test or compute CI using bootstrap.**
>
> Thank you for your insightful remark. We note that the f-GAN cannot be directly applied to obtain a useful divergence for similar reasons as the C2ST discussed in the shared response.
>
> Background: Nguyen et al. (2009) developed an M-estimation to compute the f-divergence from two samples. This estimate was used in f-GAN to replace the binary classifier. Belghazi et al (2018) developed another tighter sample-based M-estimation for KL-divergence.
>
> The sample-based M-estimation in Nguyen et al. (2009) or Belghazi et al (2018) is not directly applicable to the simulation based calibration (SBC) setting because:
> (1) the method in Nguyen et al. (2009) and f-GAN applies to IID samples, but the SBI joint draws $(\theta, y)$ and $(\tilde \theta_m, y)$ are not IID draws from two simulators since all the $\theta$ and $\tilde \theta_m$ are paired with an identical $y$, and
> (2) when the inference $q(\theta|y)$ is MCMC, there is additional auto-correlation in $\tilde \theta_m$ draws.
>
> Similar to how we can see our method as a generalization of the C2ST, it is likely that we could adapt Nguyen et al. (2009) or Belghazi et al (2018)'s KL estimate to the SBI setting. Indeed, it is the contribution of our paper to formulate the traditionally rank- and histogram-based SBC problem into a sample-based joint-space discriminative task. Our framework is flexible to incorporate the inclusion of other sample-based divergence estimates and loss functions, such as the aforementioned sample KL estimates, Wasserstein distances, and the integral probability metrics. However, we leave this as future work.

---

> > ### Comment · Reviewer_TMBR · 2023-08-14
> > **Thanks for responding**
> >
> > Thanks for responding to my question, and that clarified my misunderstandings.
> >
> > I will raise my score from 5 to 6 and vote to accept this paper.

---

### Official Review · Reviewer_eYcd · 2023-07-07

**Soundness:** 3 good
**Presentation:** 3 good
**Contribution:** 3 good
**Rating:** 7
**Confidence:** 4

**Summary:**

This work generalizes the well known Simulation-Based Calibration method to a setting where the tests on the posterior approximation consistency are based on classifiers. The paper has a nice balance between theoretical discussion (Section 3) and more practical issues (Section 4) making it easy for readers to appreciate different aspects of their proposal.

The paper is solid and tells a nice story from start to finish, but I would be tempted to point out the limitation of assessing only the quality of the posterior approximation **in average**. The authors mention this downside but don't give any insight regarding in which situations this might be a problem and when it could be OK to abandon local information.

Furthermore, it comes somewhat as a surprise the fact that the authors use a **discriminative** approach for assessing the quality of the posterior approximation, but make absolutely no mention to the well known classifier two-samples test (C2ST).

In all, I'm rather satisfied by the paper as a contribution to the SBI literature, but I'm not certain of how wide is its scope for the NeurIPS conference. I'm giving it, therefore, a score of 6 (borderline accept), and would not be surprised if in the end it was not accepted to the conference main track.

**Strengths:**

- The authors managed to recast the simulation based calibration (SBC) method in a discriminative framework. This is very nice as it helps better understanding such method and see how it compares to other approaches.
- The three theorems presented in Section 3 are of great practical utility and ensure that even though we only estimate a classifier between samples from a fixed-size dataset, we can be sure that its performance will serve as a lower bound to other theoretical quantities. In other words, the fact of actually having to abide with the "real life" setting of training a classifier from limited data is not a problem.

**Weaknesses:**

- The mathematical notation can be sometimes a bit confusing. For instance, it is not always easy to discern what is a scalar input variable to a function with what is a random variable samples from a pdf.

- There's no comment on assessing the quality of the classifier trained on calibration data and then used to build the statistical tests.

- It could have been of interest to consider more examples on simulated data to illustrate the procedure. For instance, using examples from the `sbi-benchmark` which are becoming more and more used in the simulation-based inference literature.

**Questions:**

- Why is your Figure 2 **after** the set of Figures 3-6? This is very disturbing...

- How useful it is to have a diagnostics to $q(\theta \mid y)$ which is only valid in average over $y$? Can you tell a bit more about this?

- The class imbalance in your Example 1-4 will probably make your classifier have a lot of trouble to discriminate one class from the other. Indeed, it will become more of an "anomaly detection" kind of thing than an actual classification. You don't make much comments on this but I think it would be very important to explain whether this is a difficulty or not.

- Can you add a reference to your claim in Line 82?

- Line 188: What do you mean with "classifier $c$ is good enough" ?

**Limitations:**

I've mentioned the limitations in the previous fields.

---

> ### Author Rebuttal · Authors · 2023-08-09
>
> We are grateful for your careful reading and constructive comments. We have now addressed your concerns to improve and clarify the manuscript. Please find below a detailed point-by-point response to all of your comments and questions.
>
> **Figure 2 index**
>
> Sorry for this confusion. We will reindex.
>
> **How useful it is to have a diagnostics to  which is only valid in average over?**
>
> Thank you for suggesting this important aspect! Simulation-based diagnostics are averages. Our method is consistent with the rest of the simulation based calibration (SBC) literature (e.g., the seminary papers of Cook et al. 2006 and Talt et al. 2018) in computing a measure that is an average over y. (This is often called a “global” diagnostic in the SBC literature.) SBC was originally designed to validate if *computer software* accurately draws samples. For this application, the “in-average” assessment is a feature, not a bug, especially for amortized inference. This is our main motivation for using such a measure.
>
> That said, the “local” diagnostic can be a useful goal as well: when the inference is only intended to run once, and the user is only interested in the inference given one  $y$. Our method is still relevant because KL$(p(\theta|y), q(\theta|y))=0$  if and only if $p(\theta|y) =  q(\theta|y)$ almost everywhere (Theorem 1). With modern inference routines such as MCMC or neural posteriors, we do expect that the “global divergence” could often achieve zero, which guarantees a zero local divergence with probability 1.
>
> Lastly, our classifier could be adapted to “local” calibration (last paragraph on page 9): it is enough to look at the classification performance around a small neighborhood of the observed data $y_obs$ in the simulation table. It underscores the flexibility of our framework, and we leave it for future work.
>
> **The class imbalance in your Example 1-4 will probably make your classifier have a lot of trouble to discriminate one class from the other.**
>
> We apologize for this confusion, but our novelly developed multiclass label-mapping and multiclass classifiers are always *balanced* (example 4, where we create M+1 labels from M+1 classes from each simulation draw). Hence increasing the number of inference draws (M) is not a difficulty for us. We have developed asymptotic theory and empirical validation to this big M limit.
>
> We refer to our **Share Response #3** on the detailed discussion on the balanced multiclass classifier.
>
>
> **Can you add a reference to your claim in Line 82?**
>
> Line 82 states that, when the inference is exact ($p(\theta|y) = q(\theta|y)$ almost everywhere), and we train a binary classier to distinguish $(\theta, y)$ and $(\tilde \theta, y)$, then the expected test log predictive density of a binary classifier in a 1:M imbalanced labels could be no higher than the negative binary entropy h(w) := w log w + (1 - w) log(1- w) of a Bernoulli distribution with w := 1/(M + 1). Intuitively, under the null, the best classifier (in terms of the expected log predictive density) is a Bernoulli(w) distribution. This statement is a direct consequence of our Theorem 1, Equation 8.  More detailed derivation can be found in the supplement, starting with line 589. We will clarify this point in the text.
>
>
> **Line 188: What do you mean with "classifier is good enough"?**
>
> This statement was unclear and we apologize. We will revise Line 188. What we meant was that (a) for any classifier, the divergence estimate from test data is always a lower bound of the divergence and (b) when the classifier is “good”, such classification performance is further a “good” estimate of the divergence. Here a classifier being “good” means it achieves a low test data error or equivalently a high expected log predictive density.
>
>
> **It comes as a surprise that the authors make absolutely no mention to the well known classifier two-samples test (C2ST).**
>
> It was an oversight that we did not cite the phrase “classifier two-samples test” in the related literature, for which we apologize. We will add citations to the general C2ST framework in the revision. We stress that our paper is not simply applying C2ST to SBI. We generalize C2ST by incorporating a label mapping, which allows autocorrelation and multiclass classification for the two sample tests. We would like to refer to our **Shared Response #2** for a detailed discussion of our method in relation to the original C2ST.
>
>
>  **There's no comment on assessing the quality of the classifier trained on calibration data and then used to build the statistical tests.**
>
> At a high level, the techniques used to try to make the classifier generalize well are the same as any classification problem where the goal is to generalize to test data. Our Section 4.1 gives practical recommendations on network training and feature engineering.
>
> Besides, even when the learned classifier is not optimal (as would be common in practice) the divergence we estimate is always a valid lower bound and hypothesis testing is valid. Indeed, for any classifier, the estimated divergence is always a valid lower bound of the actual divergence (Theorem 1), and the proposed hypothesis test is always valid (Theorem 4). A good classifier only helps increase the power and reduce ​​false negatives (false positives are always controlled). We do observe in finite sample examples that the power from our test is higher than others.
>
> **It could have been of interest to consider more examples of simulated data to illustrate the procedure.**
>
> Thank you for your suggestions. To showcase our method, we now add three simulation data examples from this `sbi-benchmark` repo, the Gaussian linear, the Gaussian mixture, and the simple likelihood complex posterior (SLCP) example. We run our calibration method on these three datasets with varying inferences settings and it picks up positive divergence estimates when the inference is not exact. Please see our **Shared response #1** and attached pdf file.

---

> ### Comment · Reviewer_eYcd · 2023-08-14
> **Raising my score**
>
> I thank the reviewers for their very clear answers and for adding new results from the  `sbi-benchmark`.
>
> The points which were confusing to me are no longer obscure and I will be raising my score from Weak Accept (6) to Accept (7)
>
> Best regards.

---

### Author Rebuttal · Authors · 2023-08-09

We would like to thank all the reviewers for their careful review and insightful comments. In addition to the point-by-point response we make to each reviewer individually, below we will give three shared responses, including additional experiments (in the submitted .pdf file).

### 1. Additional experiments

As suggested by reviewers, to showcase our method on a wider range of examples, we now add three simulation data examples from the `sbi-benchmark` repo:
1. the simple likelihood complex posterior (SLCP) example,
2.  the Gaussian linear model,
3. the Gaussian mixture model.

The attached pdf file summarize the experiments. Here we run our calibration method on these three datasets. In each dataset, we sample prior draws from the default prior, and run adaptive  No-U-Turn Sampler (NUTS). We want to check the quality of the sampler after a fixed number of iterations. To this end, we run the sampler with various numbers of iterations from 2 to 2000 (for each point, we use the same number of iterations for warm-up and for sampling, and the warm-up samples were thrown away. So 1000 in the $x$-axis means 2000 total MCMC iterations were run and the last 1000 were kept as the inference output), and for each number of iterations, we run our classifier calibration. The y-axis is the estimated divergence at the given MCMC iterations, and we visualize the plus and minus one standard error from our method. A positive divergence indicates a mismatch between the true posterior $p(\theta|y)$ and the inference $q(\theta|y)$, while the divergence being zero means that the posterior inference is exact, i.e., $q(\theta|y)= p(\theta|y)$ almost everywhere.

From the figures attached in the pdf file, in all three examples, we are able to detect the inference flaws and return a positive divergence estimate after a few iterations. The estimated divergence also does converge to zero for more iterations where we expect inference to be near exact.

### 2. [C2ST] Our paper is not simply applying the classifier two-sample test (C2ST) to SBI calibration. We generalize the classifier two-sample test.

Indeed, using a classifier to perform a two-sample test is not a new idea. We did not intend to imply this but we acknowledge that the paper as submitted does not make this as clear as it should. That said, the classifier two-sample test is not directly applicable to simulation based calibration (SBC) due to four barriers:
1. In the past of SBC, it was not clear what space to run the classifier on, i.e. should it include parameters $\theta$, data $y$, or the joint, or include likelihood values, nor to interpret results as a divergence between the true and inferred posteriors.
2. More importantly, the classifier two-sample test only works with IID examples (there are two distributions P and Q, and we have IID observations from P and Q respectively.)  The SBI joint draws are not IID because the simulation table contains shared y. Think about our binary classification scheme, the example from class 0 is $(\theta, y)$, and the examples from class 1 is $(\tilde \theta_1,  y), … (\tilde \theta_M,  y)$; they all share a same y component.
3. When the inference is done MCMC, there is additional auto-correlation in $\tilde y$, which further violates the C2ST requirement.
4. For two-sample tests, C2ST only creates binary labels and may perform poorly with imbalanced classification. Using naive binary classification in SBI will face a 1:S imbalancement.

Our paper solves these issues by developing a general label mapping framework:
- For (1), our paper formulates the SBC problem into a sample-based joint-space discriminative task for the first time. We develop the relevant theory for the SBI calibration and prove its relevance to posterior divergence. We also make use of additional information like log-likelihoods in a unified framework.
- For (2), our theorem 1 and 9 extend the traditional IID two sample tests, and allows a sequential sampling.
- For (3) and (4), we extend the straightforward binary classification to a general framework “the label mapping” (line 132). The traditional binary C2ST is now a special case of our framework (example). In contrast, the multiclass classifier (example 4 in Section 2) is always balanced and allows the examination of auto-correlated samples (equation 12).

### 3. [balanced classifier] The naive binary classifier can suffer from imbalanced labels, but our novel multiclass classifier always has perfect label balances.
Our novelly-developed multiclass classifier (example 4 in Section 2) is always perfectly *balanced*: each simulation run creates (M+1) examples from (M+1) classes; one from each. See Page 3 for an illustration table. Our theorem 3 proves that the multiclass classification divergence converges to the meaningful KL divergence between p and q and derives the convergence rate, under the limit $M \to \infty$, the simulation in which the naive binary classification is infinitely imbalanced.

That being said, the binary classifier might be more intuitive to the users than the multiclass one, and we still support the use of binary classifiers in our calibration. In our experiment, we find that even under highly imbalanced binary labeling (M=1000), with appropriate sample reweighting (Theorem 8 in Appendix), the binary classifier is still able to detect the inference flaws and output accurate divergence estimates.

---

> ### Comment · Reviewer_TMBR · 2023-08-14
> **(f)-Divergence View has been Studied in SBI**
>
> Thanks for the response.
>
> > In the past of SBC, it was not clear what space to run the classifier on ... nor to interpret results as a divergence between the true and inferred posteriors.
>
> First, the second part of this claim is not true. See,
>
> Section 2.1, Variational Inference for Simulation-based Inference, ICLR 2022, where authors clearly formulated the SBI problem as a divergence minimization problem between true and inferred posterior.
>
> Second, I do not understand the first half of the sentence. The binary classifier has been studied to minimize such type of divergence (in an adversarial fashion) in the space of both theta and y, see
>
> GATSBI: Generalized Adversarial Training for SBI, ICLR 2022.
>
> thus I do not understand why "it was not clear what space to run the classifier"? Please specify.
>
> -------------------------------------------------------------------------------------------------------------------------------------------------
> **Revision**:
>
> Sorry, I noticed SBC stands for SB "Calibration", not "Computation", thus authors claims **are valid**.
>
> Nonetheless, these two papers I mentioned above seem to be relevant as they both considers computing divergence measure in SBI problems.

---

> > ### Author Response · Authors · 2023-08-15
> >
> > Thank you for this comment and we apologize for the confusion, where we were only referring to the divergence estimation and interpretation in the simulation-based calibration (SBC). The GATSBI paper developed adversarial inference for SBI via a binary classifier and a one-to-one balanced generator. It was an overlook that we did not discuss this paper and we apologize.

---

### Comment · Area_Chair_nLF8 · 2023-08-15
**Author-reviewer discussion**

Dear all,

The author-reviewer discussion period has now started. It will continue for one more week, until August 21.

@authors: Please respond to the comments or questions reviewers may further have. Remain short and to the point.

@reviewers: Please read the author's responses and ask any further questions you may have. To facilitate the decision by the end of the process, please also acknowledge that you have read the responses and indicate whether you want to update your evaluation.

- You can update your evaluation positively (if you are satisfied with the responses) or negatively (if you are not satisfied with the responses or share other reviewers' concerns). Please note that major changes are a reason for rejection.
- You can also keep your evaluation unchanged. In this case, please indicate that you have read the responses and that you do not have any further comments.

Best regards,
The AC

---

### Decision · Program_Chairs · 2023-09-21

**Decision:**

Accept (poster)

**Comment:**

The reviewers unanimously recommend acceptance (7-6-7-7). Several minor issues have been raised and discussed during the author-reviewer discussion period. The authors are encouraged to take this feedback into account in the final version of the paper.